# TermPicks: A century of Greenland glacier terminus data for use in scientific and machine learning applications

Sophie Goliber[1,2], Taryn Black[3,4], Ginny Catania[1,2], James M. Lea[5], Helene Olsen[2], Daniel Cheng[6], Suzanne Bevan[7], Anders Bjørk[8], Charlie Bunce[9,10], Stephen Brough[5], J. Rachel Carr[9], Tom Cowton[11], Alex Gardner[12], Dominik Fahrner[5,13], Emily Hill[14], Ian Joughin[4], Niels Korsgaard[15], Adrian Luckman[7], Twila Moon[16], Tavi Murray[7], Andrew Sole[17], Michael Wood[12], and Enze Zhang[18]

[1]Department of Geological Sciences, University of Texas at Austin, Austin, TX, USA
[2]Institute for Geophysics, University of Texas at Austin, Austin, TX, USA
[3]Department of Earth and Space Sciences, University of Washington, Seattle, WA, USA
[4]Polar Science Center, Applied Physics Laboratory, University of Washington, Seattle, WA, USA
[5]Department of Geography and Planning, University of Liverpool, Liverpool, UK
[6]University of California at Irvine, Irvine, CA, USA
[7]Geography Department, College of Science, Swansea University, Swansea, UK
[8]Department of Geosciences and Natural Resource Management, University of Copenhagen, Copenhagen, Denmark
[9]School of Geography, Politics and Sociology, Newcastle University, Newcastle-Upon-Tyne, UK
[10]School of Geosciences, University of Edinburgh, Edinburgh, UK
[11]School of Geography and Sustainable Development, University of St Andrews, UK
[12]Jet Propulsion Laboratory, California Institute of Technology, Pasadena, CA, USA
[13]Institute for Risk and Uncertainty, University of Liverpool, Liverpool, UK
[14]Department of Geography and Environmental Sciences, University of Northumbria, Newcastle upon Tyne, UK
[15]The Geological Survey of Denmark and Greenland, Østervoldgade 10, 1350 København K, Denmark
[16]National Snow and Ice Data Center, Cooperative Institute for Research in Environmental Sciences, University of Colorado Boulder, Boulder, CO, USA
[17]Department of Geography, University of Sheffield, Sheffield, UK
[18]Earth System Science Programme, The Chinese University of Hong Kong, Hong Kong, China

**Correspondence:** Sophie Goliber (sgoliber@utexas.edu)

**Abstract.** Marine-terminating outlet glacier terminus traces, mapped from satellite and aerial imagery, have been used extensively in understanding how outlet glaciers adjust to climate change variability over a range of time scales. Numerous studies have digitized termini manually, but this process is labor intensive, and no consistent approach exists. A lack of coordination leads to duplication of efforts, particularly for Greenland, which is a major scientific research focus. At the same time, machine learning techniques are rapidly making progress in their ability to automate accurate extraction of glacier termini, with promising developments across a number of optical and SAR satellite sensors. These techniques rely on high quality, manually digitized terminus traces to be used as training data for robust automatic traces. Here we present a database of manually digitized terminus traces for machine learning and scientific applications. These data have been collected, cleaned, assigned with appropriate metadata including image scenes, and compiled so they can be easily accessed by scientists. The TermPicks data set includes 39,060 individual terminus traces for 278 glaciers with a mean of $136\pm190$ and median of 93 of traces per glacier. Across all glaciers, 32,567 dates have been digitized, of which 4,467 have traces from more than one author and there

is a duplication rate of 17%. We find a median error of ∼100 m among manually-traced termini. Most traces are obtained after 1999, when Landsat 7 was launched. We also provide an overview of an updated version of The Google Earth Engine Digitization Tool (GEEDiT), which has been developed specifically for future manual picking of the Greenland Ice Sheet.

# 1   Introduction

Since the 1980s, the Greenland Ice Sheet (GrIS) has been in negative mass balance due to increased surface melt and ice discharge (Mouginot et al., 2019; Enderlin et al., 2014) with projected increases in sea level of 5 to 33 cm by 2100 from Greenland alone (Aschwanden et al., 2019; Goelzer et al., 2020). Long-term historical trends in ice sheet mass loss show that approximately 50% of the total mass loss since the ∼1990s is from ice dynamics alone, via fast-moving outlet glaciers that drain into to the ocean (Enderlin et al., 2014; Mouginot et al., 2019; King et al., 2020). In part, this acceleration in dynamic loss may have been triggered by a warming climate (atmosphere and ocean) that induces sudden rapid retreat of outlet glacier termini (Wood et al., 2021; King et al., 2020). Observations of glacier retreat, however, show a high degree of heterogeneity in the magnitude, timing, and temporal patterns of this retreat across the ice sheet (Moon and Joughin, 2008; Catania et al., 2018; Murray et al., 2015a; Carr et al., 2017; Fahrner et al., 2021), which complicates our understanding of future mass change from outlet glaciers. This suggests that knowledge of past terminus change, and the potential for future terminus change, is critical for accurate forecasting of the GrIS contribution to sea level rise (*e.g.* Felikson et al., 2017; Aschwanden et al., 2019; Slater et al., 2019).

Glacier termini have long been an indicator of climate change and terminus change data have been used to understand a range of processes over multiple time scales (*e.g.* Warren and Glasser, 1992; Warren, 1991; McNabb and Hock, 2014; Moon et al., 2015; Cook et al., 2005; Howat et al., 2008; Howat and Eddy, 2011). On the long-term (>annual), terminus records are used to inform the timing of, regional patterns within, and climate controls on marine-terminating glacier retreat (Murray et al., 2015b; Catania et al., 2018; Hill et al., 2018; Bunce et al., 2018; Howat and Eddy, 2011; Wood et al., 2021; King et al., 2020; Fahrner et al., 2021; Black and Joughin, 2022). Outlet glaciers can also change at sub-annual timescales and examination of terminus change on shorter time scales (∼seasonal) aids interpretation of the specific environmental and glaciological processes that influence glaciers (Fried et al., 2018; Moon et al., 2015; Schild and Hamilton, 2013; Cassotto et al., 2015; Ritchie et al., 2008; Howat et al., 2010; Carr et al., 2014; Moon et al., 2014, 2015; Brough et al., 2019; Kehrl et al., 2017; Bevan et al., 2019). Such studies are valuable because glacier termini respond to a diverse set of mechanisms related to the geometry of the glacier-fjord system, inland ice dynamics, and the strength of climate forcing (Moon and Joughin, 2008; Carr et al., 2017; Catania et al., 2018; Bunce et al., 2018; Porter et al., 2018). However, determining the variables controlling seasonal variations can be difficult because changes in the climate system occur simultaneously (e.g. Cowton et al., 2018; Fahrner et al., 2021; Wood et al., 2021). Recent work suggests that the shape of the terminus trace and how it evolves over time may provide additional information about the nature of processes dominating any given glacier (Fried et al., 2018; Chauché et al., 2014). Such studies demonstrate the need for detailed tracing of the full terminus width (in map-view) at as high a temporal resolution as possible.

Numerous studies have digitized termini manually (Table 1) for use in interpreting glacier dynamics in response to climate

variability; however, the lack of coordination across these studies has resulted in duplicated data and heterogeneity in terms of format, quality, method, location, temporal coverage, and availability. Such factors limit the utility of terminus data to future researchers. In addition, manually picking glacier termini is a laborious process. For example, the data set from Catania et al. (2018) used the entire Landsat record to digitize 15 glaciers in central West Greenland and the authors estimate that it took 3 undergraduate researchers nearly 2 summers working 15 hours a week each to download imagery and digitize the full width

of the terminus, or approximately 48 hours per glacier. Rapidly replacing manual-picking are machine learning techniques, which have recently been developed for automated extraction of glacier termini across a number of satellite sensors (*e.g.* Mohajerani et al., 2019; Baumhoer et al., 2019; Cheng et al., 2020; Zhang et al., 2021). Manually-digitized data are still needed for validation of machine learning methods and as training data. For example, methods using over 1500 training data inputs result in classification in ∼94% of detectable images, under ideal conditions (Cheng et al., 2020). Further, machine

learning methods fail in images where ice conditions do not permit easy delineation of the terminus (*e.g.* mélange-choked fjords, shadowed termini, etc.) and therefore manually-digitized termini will still be needed until machine learning algorithms improve. Importantly, future satellite missions imaging the polar regions are expected to continue for the foreseeable future, suggesting an ongoing need to coordinate terminus data in addition to other important glaciological observations that are highly coordinated (*e.g.* velocity and elevation). Here we present the most complete set of manually digitized terminus data for

Greenland's outlet glaciers, re-processed for use in machine learning methods and scientific analysis. Data have been cleaned, associated with appropriate metadata where possible, and the metadata normalized so they can be easily accessed by scientists.

## 2 Methods

### 2.1 Input data

Terminus traces were collected through email requests to authors who had published papers that made use of such data, or

65 taken from publicly available online databases (Table 1). Since there was no open call for data submission, there may be other sources of terminus trace data that are available and/or unpublished. Authors used a range of image sources (Table 2), but the bulk (∼70%) of terminus traces originate from Landsat images. Collectively, we refer to these collected data as input data to differentiate these data from the output (cleaned, reformatted) training data generated.

All data were provided in ESRI shapefile format (Figure 1) with the bulk of data provided as polylines and a smaller volume

of data provided as polygons or polygon-boxes. In these latter cases, the polygons were cropped at the terminus and converted into polylines. All glacier terminus traces were exported into a single ESRI line shapefile format consistent with file formats typically used in machine learning techniques. All shapefiles were re-projected into NSIDC Sea Ice Polar Stereographic North (EPSG:3413).

Glacier termini were commonly traced by importing geographically-rectified images into GIS software (*e.g.* ArcGIS, ENVI,

and QGIS) and manually-digitizing the ice-ocean boundary (terminus). Authors used a range of methods for tracing termini

including picking the full width or variations on the Box methods. Box methods consist of using a fix-width rectilinear or curvilinear box along the length of a fjord tracing the terminus within those bounds (for a description of these methods see Lea et al., 2014). For consistency in data format, we excluded termini that were identified with only a center point (*e.g.* King et al., 2020) because these data do not cover the entire width of termini. Individual terminus trace files are largely indistinguishable between authors, with the exception of those who used the box method for picking the terminus, since this method often produces terminus traces that are truncated before they reach the fjord wall. Across all authors, terminus traces have an average of 23 vertices per kilometer with a median of 10 vertices per kilometer.

## 2.2 Glacier identification

As the GrIS has several hundred marine-terminating glaciers, proper identification of glaciers is important for data management. Several prior authors have produced identification files (ID files) for GrIS glaciers including Moon and Joughin (2008) (Moon IDs) who created a glacier ID file by identifying all non-stagnant glaciers that terminate in the ocean with terminus widths of roughly 1.5 km or greater. The Moon IDs identify 239 glaciers that are assigned a numerical ID, including 6 ice cap glaciers that are marine-terminating. We received terminus traces for 278 glaciers but subsequently identified 282 glaciers by including all glaciers with a Moon and Joughin (2008) ID and additional glaciers with the following criteria; 1) surface speeds >50 m/yr, 2) grounding lines below sea-level as determined from the BedMachineV3 bed topographic product (Morlighem et al., 2017), and 3) termini greater than or equal to 1 km in width. We excluded terminus traces where only one pick was available for the glacier over all authors as well as land-terminating glaciers (Mouginot et al., 2019). Using this new ID system, here termed TermPicks ID, we assigned glacier IDs to each glacier in our database (Figure 1).

Our TermPicks ID file maintains consistency with the Moon IDs by including the corresponding Moon ID with the TermPicks ID file. We also includ other information in the TermPicks ID file that is relevant for wide community use, including outlet glacier flux gates identified by Mankoff et al. (2019) and glacier naming schemes catalogued by Bjørk et al. (2015) in an ESRI multipoint shapefile so the data can be easily referenced with other data sets.

## 2.3 Data cleaning

The number of terminus traces included in an input shapefile varied across the input data. Some authors represented multiple dates per glacier within each shapefile while others included single dates per glacier for each shapefile. Our output data merged all terminus traces for all dates together into one shapefile and so input data were re-processed to fit into this format. Some authors included multiple glaciers per date for a shapefile, particularly when glaciers were adjacent to one another. Where possible, these shapefiles were manually split into traces representing separate glaciers, consistent with our output data format (Figure 2c). This was accomplished using the MEaSUREs Greenland Ice Mapping Project (GrIMP) 2000 Image Mosaic (Howat et al., 2014; Howat, 2018) for glaciers to be properly sorted along fjord wall boundaries or ice stream where appropriate. Traces were also clipped using the GrIMP ice mask in order to remove fjord wall traces (Howat et al., 2014). The mask was extended where it did not intersect earlier traces.

Traces that were digitized using the box methods were not interpolated to the fjord wall. In many cases, the box spans nearly the entire width of the fjord, but several datasets use boxes that are much smaller than the width of the fjord (Figure 2a). The lack of data at the edges of glacier termini may lead to differences in total retreat using these data compared to other data (Lea et al., 2014). Thus, terminus traces digitized using the box method are flagged in the metadata (Table 3).

## 2.4 Metadata creation

Consistent and uniform metadata are critical to the use of training data in machine learning and scientific studies. Feature extraction using image segmentation techniques rely on accurate attribution of training data to the correct time, location and satellite image used for terminus tracing. Input data used for TermPicks suffered from a lack of consistency in the metadata, such as date format, author and satellite identification, image ID, and digitization techniques. Here we describe the metadata format for the output TermPicks data set (Figure 1). The TermPicks metadata format was chosen to be consistent with the largest archive of machine-digitized terminus traces from Cheng et al. (2020), known as CALFIN. For example, CALFIN includes the date, quality flags, satellite sensor and image ID, all of which are important for machine learning. Figure 3 shows examples of the metadata structure for the data.

**Date columns:** The Date column represents the acquisition time for the image used to digitize the terminus for that trace. There are 4 additional columns for year, month, day and decimal date. The Date column is a string and the format is "YYYY-MM-DD". Year, month, and day are integers. If a trace included only year information, the date column format is "YYYY-00-00".

**Satellite:** Satellite refers to the original sensor or satellite that produce an image used to digitize the terminus. This information was taken from existing attribute tables or file names from the input data and was used to determine the image ID where possible. The names used are in listed in Table 2.

**Author:** All people contributing traces have been listed as authors in this paper. Included in the metadata is the Author identifier connected to a specific citation using the data provided. We also provide a code block in the code repository to produce citations for the authors of terminus traces that are used in data downloads. This allows for proper attribution to the correct author depending on the location and time span of data downloaded. In the data set, the author 'TermPicks' refers to terminus traces produced with TermPicks GEEDiT, but are not published elsewhere (Appendix C).

**Image ID:** Image ID refers to the image scene identifiers for the original image used to digitize the individual glacier trace. This corresponds directly to the sensor. For example, a Landsat Product ID is an example of an image ID. Certain images (*e.g.* some aerial images) were used to digitize multiple traces. The image ID includes information on the date and location for the original image. This may be listed as a file name that the original author used and may store locally (Figure 3; Glacier 291) or an image ID from a different satellite (*e.g.* Sentinel-1 product folder name). If an author included an image ID, the text was

kept the same in case users need to contact the original author for image access.

**Glacier IDs:** The Glacier ID refers to the TermPicks glacier ID scheme that was created for this project (described in section 2.2).

**Center X and Y:** A centroid point was created for each trace in WGS 84 (EPSG:4326) so that the TermPicks data can be easily referenced with other data sets.

**Quality flag:** Quality flagging is used to identify and classify traces that may have issues leading to sources of error. This quality flagging schemed was created in conjunction with Cheng et al. (2020) to enable data synthesis between our data and machine-generated terminus traces. We assign a prefix 'X' for all data defining whether the trace was created automatically or manually, with X=0 for TermPicks data and X=1 for CALFIN data, or any machine-generated terminus traces that may be included in the future. In addition, traces can have multiple quality flags. We follow the quality flag scheme in Table 3. In this scheme, flags are assigned if there are no issues with the terminus trace (X0), if there is uncertainty in the trace due to environmental or image issues, for example clouds partially obscuring the terminus (X1), if the trace was supplemented (two images were used to digitize the terminus) (X2), if the trace was digitized with the Landsat 7 sensor when the Scan Line Corrector was off (X3), if the trace was digitized using the box method and is thus incomplete (X4), if the image ID was automatically assigned because of lack of information provided in the input metadata (X5). The X1 and X2 flags are only used if the trace author indicated this information, and so many traces will not include these flags. If there are multiple flags, they are separated by commas (Figure 3; Glacier 278).

## 2.5   Landsat image scene identifiers

Satellite image scene identifiers (Image IDs) are useful to find the original image from which a glacier terminus was digitized, which is a requirement for these data to be useful for machine learning. Including image IDs is also useful in cases where scientists want to explore other features in the scene at the time of a terminus trace (*e.g.* iceberg distribution, sediment plume occurrence). These were provided in very few of the input data sets. Where no image ID was available, Landsat scene identification is assigned to terminus traces that were originally digitized using Landsat data. Scenes were assigned by geolocating a Path/Row from the Worldwide Reference Systems (WRS-1 for Landsat 1-3; WRS-2 for Landsat 4 onward) that is closest to the terminus trace, then searching by date using Google Cloud Services. As Landsat scenes are freely available for Level-1 data on Google Cloud Services and most ($\sim 70\%$) of the data are derived from Landsat images, only terminus traces that were known to be digitized with Landsat data are assigned IDs (Figure 1). Some glaciers share multiple overlapping Landsat Path/Row combinations resulting in some terminus traces having two scenes assigned. In these cases, both image IDs are appended to the metadata. Glaciers with automatically assigned image IDs have the quality flag of 05 (Figure 3; Glacier 3). Further, some

terminus traces did not have dates that corresponded to an image ID from Google Cloud Services and were not assigned an image ID.

## 2.6  Calculation of terminus change and variability

In addition to providing manually-digitized terminus traces for glaciers in Greenland, we also computed terminus position change. As many previous studies have already published on terminus change over time, we provide these estimates largely as a check on our data set. We compute terminus position in two ways. First, we calculate terminus position using a method developed in Catania et al. (2018) where equally-spaced points along each terminus trace are projected to the nearest location along the glacier centerline. The average position of all projected points on the centerline thus becomes the average position of the glacier terminus for that date of the terminus trace. We call this the Interpolation Method. The Interpolation Method is most accurate when the glacier traces are all approximately the same length (*i.e.* not a mixture of full-width and box-method termini). Second, we calculate the fluctuation in terminus position simply by taking the point where the terminus intersects the centerline of each glacier following King et al. (2020), here named the Centerline Method. Traces that were missing day and month information were assumed to have a timing of mid-year. Retreat rates were then calculated by taking the distance between each of these terminus positions over time. We use centerlines from Murray et al. (2015a) where available for the glaciers in our database. Remaining centerlines were manually mapped from the MEaSUREs Greenland Ice Mapping Project (GrIMP) 2000 Image Mosaic (Howat et al., 2014; Howat, 2018) through the center of the glacier and the terminus traces.

We also computed the terminus seasonality as a measure of the total variation in the terminus position over the annual cycle. This is quantified using the standard deviation of the difference between raw terminus position data and smoothed terminus position data from the centerline following Catania et al. (2018). We estimated seasonality for glaciers in years where there are terminus traces in at least three unique months.

Finally, we calculated the terminus sinuosity as a way to characterize the shape of the terminus, as the sinuosity quantifies how much the terminus deviates from a straight line. Sinuosity is classically used in river morphology to describe map-view morphological changes in river channel patterns and is ratio of along-channel length to valley length (Schumm, 1985; Montgomery and Bierman, 2019). Here, terminus sinuosity is measured as the length of the terminus divided by the straight line distance between the terminus end points. Sinuosity of rivers depends on river valley geology with typical values between 1 and 3 (Schumm, 1985), however, we do not expect glacier termini to exceed a sinuosity of 2 (*i.e.* the terminus will be less than twice the length of the distance across the fjord) because calving will likely occur for the parts of the terminus that are extremely anomalous. Increased sinuosity of glacier termini may be associated with crenulated terminus morphology that is thought to result from localized terminus melt as a result of buoyancy-driven plumes (Chauché et al., 2014; Fried et al., 2018); however, a smooth but highly concave terminus may also have a high sinuosity. Low sinuosity termini may be associated with glaciers that calve via full-thickness calving events, causing fjord-width step changes in the terminus position with each calving event (Fried et al., 2018; James et al.). While additional metrics of the geometry (*e.g.*, curvature) may be necessary

to completely describe the morphology of glacier termini, the change in sinuosity in time may reveal differences in processes affecting a single glacier.

## 2.7 Error Estimation

Terminus traces from different authors on the same date do not necessarily align with each other, and so we quantified the difference between these traces. As a metric of error between data sets, we calculated the Hausdorff distance (commonly used in pattern recognition), the greatest minimum distance between two lines (Huttenlocher et al., 1993). A larger Hausdorff distance indicates two lines are less similar to each other; however, large Hausdorff distances could also indicate that two otherwise identical lines have different endpoints (different lengths). To avoid this latter issue, we trimmed each terminus trace to a glacier reference box, modified from those used by Moon and Joughin (2008), before computing Hausdorff distances. We also excluded traces that did not span the width of these glacier boxes. Excluding short traces reduced the dataset to 25,355 (65% of the original TermPicks dataset). Then, we calculated the Hausdorff distance between every pair of traces for traces that were digitized at the same glacier and on the same date by multiple authors. We identify 2,671 individual instances where multiple authors digitized a glacier on the same date (sometimes more than two authors). This resulted in a total of 5,748 duplicated traces.

# 3 Results

The TermPicks data set includes 39,060 individual terminus traces for 278 glaciers with a mean and median number of traces per glacier of 136±190 and 93, respectively. However, trace count varies depending on author interest in a specific glacier or region of glaciers (Figure 4). Across all glaciers, 32,567 dates have been digitized, of which 4,467 have traces from more than one author. This represents duplicated efforts of ~17% of the input data. Traces extend back to 1916 for a small number of glaciers but the greatest number of traces are obtained between 2000 and 2017 (Figure 5). See supplemental material for information on individual glacier coverage and statistics (Figures A9, A10, A11) as well as access to a kmz file that can be viewed in Google Earth that produces a quick look at location and coverage for each glacier.

## 3.1 Terminus change and variability

The retreat time-series using the Interpolation method reveals small errors that are present as anomalous spikes in the retreat record, possibly due to traces that have different endpoints (*e.g.*, Figure 6). Centerline retreat as an average over each decade of the observational record (1940-2010 where sufficient data permit) shows regional patterns of retreat in the before 1990 and more ubiquitous retreat after 1990 (Figure 7). Glacier terminus seasonality varies over time and space. Out of the 19 authors in our data set, 10 are able to resolve a seasonal signal for at least one glacier for at least one year (Figure 8). The Catania data are able to resolve seasonal signals across the longest time period (1985-2019), however this is only for 15 glaciers. The Murray data set resolves seasonality for 199 glaciers but only between 2000-2009. In contrast, the TermPicks data set resolves

seasonality for the most glaciers (n = 221) at different levels of completeness over the longest period of time (1985-2019). For example, Glacier 116 has traces from 7 authors (Figure 9), allowing us to examine changes in seasonality from over ∼35 years between 1986 and 2017. In contrast, the data from Murray only resolve seasonality for Glacier 116 for 8 years between 2000 and 2008. Finally, we find increases in the amplitude of terminus seasonality during periods of terminus retreat for all three of our example glaciers (Figure 9).

We calculate the sinuosity of Kangerdlugssup Sermerssua (Glacier 291) and Sermeq Silarleq (Glacier 288) between 1990 and 2020, as there is the highest density of traces after 1990 (Figure 10). Terminus sinuosity is found to vary generally between values of 1 (straight across) to 2 (highly sinuous). We examine two examples with different retreat histories. Glacier 291 is a stable glacier over the observational time period and has a similarly stable sinuosity with a mean of 1.43 ± 0.12 between 1990-2020 (Figure 10). In contrast, Glacier 288 undergoes a two-stage retreat beginning in 1998 with a slower paced stage of retreat until ∼2010 when retreat accelerated through to today. This glacier has a mean sinuosity of 1.35 ± 0.17, however we observe that the slower period of retreat is tied to a period of increased terminus sinuosity of 1.41 ± 0.18, (Figure 10), while the period of more rapid retreat experiences a decrease in terminus sinuosity to values of 1.29 ± 0.13 (Figure 10).

## 3.2    Spatial and temporal bias

Heatmaps of the output data demonstrate the temporal coverage and frequency of the data. We present heatmaps for both regional groups of glaciers (Figure 5) and individually for each glacier (Figures A9, A10, A11). These figures demonstrate that terminus data availability is intimately tied to Landsat image acquisition. A combination of U.S.-centric acquisition strategies, ground station coverage, and limitations on data transmission and duty cycles meant that much of the world did not have regular repeat Landsat coverage until 2013 with the launch of Landsat 8, which follows a continental acquisition strategy (Wulder et al., 2016). Further, the failure of Landsat 6 upon launch in October of 1993 meant that imagery was only obtained in a limited capacity (via extension of the Landsat 5 satellite) until the successful launch of Landsat 7 in 1999, when we observe an increase in terminus trace data (Figure 5). We further compute the percentage of terminus traces for a given glacier compared to all available Landsat images that cover any particular glacier (see Figure 11 for four examples) in order to examine the completeness of the terminus data for all glaciers. All glaciers have an individual coverage figure that is contained in our Google Earth file (Supplementary Information). From this analysis we find that Sermeq Silarleq (ID 288) has traces from 33.1% of all available Landsat images (including cloudy images), the most of any glacier in our data set. However, on average only 5.8% of available Landsat images have been manually traced per glacier.

Regional differences in data availability also exist (Figures 4 and 5). Higher latitude glaciers experience more frequent coverage by satellite image sensors than lower latitude glaciers due to increased scene overlap at high latitudes (*e.g.* Figure 11b after 2013). In Southwest Greenland, there are fewer traces simply due to the lack of marine-terminating glaciers in this region, which is primarily drained through land-terminating ice. There are also fewer overall traces in North and Northeast Greenland than Central West Greenland, a region with a similar number of glaciers, potentially due to less interest in tracing in North and Northeast Greenland (Figure 4). The densest coverage is in Central West and Northwest Greenland (IDs 279 to 3)

where nearly every available image from Landsat and other sensors were traced (Catania et al., 2018) to create as complete a
record as possible of regional glacier change. Other glaciers of interest include Helheim, Kangerlussuaq, and Sermeq Kujalleq
(Jakobshavn; IDs 181, 152, and 278), which also have dense coverage.

## 3.3 Error in manual digitization

The overall median error between pairs in this reduced dataset is 107 m, which is comparable to that obtained in most machine
learning studies when comparing machine-traced termini to manually-traced termini (Cheng et al., 2020). The median error
between any given pair of authors varies with the greatest median error (7,350 m) between Cheng and Hill, and the least median
error (58.6 m) between Fahrner and TermPicks (Figure 12). The magnitude of errors are not necessarily due to inaccurate
digitisation by authors, but can be explained by Hill and other authors focusing on northern glaciers (which can be difficult
to trace due to the presence of near-terminus crevasses), and Fahrner focusing on late summer observations where the glacier
margin is often most clear. The mean and median of the median errors for each author are presented in Table 4, and there was
no clear distinction in error based on methodology used (box vs. full-width tracing). Traces with >500 m error between traces
were manually checked for errors (220 traces). If two traces were on the same date but the trace was not equivalent (*e.g.* the
trace did not appear to be from the same front), then the trace with more complete metadata (*e.g.* includes the original image
ID) was kept. If a trace had 3 authors and one was not equivalent, it was removed. Only 0.4% of total traces were removed
from the data set through this manual checking. In some cases, there are glaciers that have higher errors than other glaciers
(*e.g.* IDs 39, 73, 86, 99, 100, 101) due to the fact that they appear to have highly fractured ice tongues and they develop long,
linear cracks that authors may or may not trace in their entirety.

Termini traced with different methods or widths of the glacier may have some systemic differences in terminus retreat
over time (Lea et al., 2014). For example, Figure 6 shows Glacier 152 (Kangerlussuaq Gletsjer) on 8/11/2006. This date was
digitized by 3 separate authors (Bunce, Cheng, and ESA) at different extents of the glacier front. When the Interpolation
method is used, there is a 0.5 km difference in terminus position change because the end points for each trace are different.
Bunce and Cheng will show a higher retreat compared to ESA because the Interpolation method accounts for the entire width
of the glacier. Therefore the mean positions of the Bunce and Cheng traces will be further up-glacier as they do not include
the lateral tails seen in the ESA trace. While there is no large scale difference between retreats calculated from the box method
versus full width traces, users of these data should be aware of this potential misfit between traces based on end points. For
example, Bunce traces use the box method while Cheng traces uses the full width method; however, they both end before the
fjord wall. Glacier 152 has dead ice on its northern margin and, as shown in the image, the scan line errors in the Landsat 7
imagery block some of the ice, so some authors may or may not digitize the entire front for numerous reasons.

# 4 Discussion

This is the first published study of manually-traced Greenland-specific marine terminating glacier traces with consistent meta-data and formatting across multiple data sets from different authors. Glacier terminus traces have been a staple indicator of glacier change for decades (*e.g.* Weidick, 1958; Higgins, 1990; Warren and Glasser, 1992; Murray et al., 2015a). From this paper alone, 22 sources have digitized and interpreted terminus positions in Greenland, with many more using these data to aid interpretation of GrIS change. However, all of these efforts have happened independently, with duplicate efforts and lack of consistency across data format and accessibility. For example, Figure 13 shows a time-series of Glacier 116 (F. Graae Gletscher) with author labels for each trace. This figure demonstrates the utility of combining data sources, which enables a more complete view of terminus change at this glacier than any previously-published individual study. We find similar ice-sheet wide retreat patterns as previously published sources. For example, total retreat for 2000-2010 is ∼252 km in 225 glaciers (Figure 7), which is comparable to Murray et al. (2015a) who found ∼267 km in 199 glaciers. We find the greatest retreats occur from 1990-2010 (Figure 7) similar to Wood et al. (2021) and Fahrner et al. (2021). Finally, we find a rapid increase in retreat beginning in 1990s-2000s (Figure 7), similar to Carr et al. (2017), King et al. (2020) and Fahrner et al. (2021). While we recognize that not every glacier has a complete time series or the ability to resolve seasonal changes in terminus position over all years and there remain limitations in drawing large scale conclusions on retreat patterns with these data alone, we find increases in the amplitude of terminus seasonality during periods of terminus retreat (Figure 9). This may be related to the changes in fjord geometry that glaciers experience as the terminus retreats through overdeepenings.

An additional value of the TermPicks data set is that it provides map-view trace data, not just centerline data, thus informing on morphological changes to the terminus over time. We explore the value of this through examination of the terminus sinuosity, but other measures (*e.g.*, terminus curvature) may also be valuable in contextualizing terminus morphology. While the mean sinuosity for Glaciers 288 and 291 (Figure 10), are similar, we find variations in sinuosity for the glacier that experienced large scale retreat (Glacier 288) compared to the one that has remained stable over the observational period (Glacier 291). Glacier 291 is known to have a terminus that is dominated by plume-driven melting (Fried et al., 2015, 2018; Jackson et al., 2017), and so we might anticipate increased sinuosity related to local melting associated with these plumes (Chauché et al., 2014; Fried et al.). In contrast to this, the terminus of Glacier 288 begins with a relatively low sinuosity, then during the period of slow retreat (1998-2010) experiences an increase in sinuosity (Figure 10) suggesting that this glacier may also have experienced enhance terminus melting due to subglacial discharge plumes during this time. Subsequently, Glacier 288 experiences a period of more rapid retreat as the glacier terminus moves into an overdeepened portion of the bed. Here, sinuosity decreases and terminus change is dominated by full-thickness calving (Fried et al., 2018).

Although machine-enabled terminus tracing has made great strides in the past few years, there will be a continued need for manually-tracing glacier termini. This is because certain environmental conditions, such as heavy shadows, cloud cover, ice mélange, and low solar illumination, make it difficult for current machine learning algorithms to accurately trace all available images. The data provided here will aid improvements in machine learning that will ultimately reduce the need for future manual tracing. Ideally, machine and manual-tracing efforts would work in concert, with data gaps or large errors reported

by machine learning quickly identifying where need is the greatest for the manual-tracing team. For example, both the data presented here and the data in CALFIN (Cheng et al., 2020) are not extended beyond 2020 and there is no funding in place to provide continued coordinated (between machine- and manual-traced authors) updates to terminus positions in the future.

Coordinated effort between machine- and manual-tracing teams is warranted to ensure regular delivery of future data, given its importance to the wider scientific community.

Until fully-automated, frequently-updated and publicly available terminus traces are available for Greenland and elsewhere, we anticipate that authors will continue to manually-trace in studies that are spatially or temporally limited. Ideally, future efforts would occur in conjunction with our work, producing data with similar format, metadata, and visibility. To that end,

we recommend the use of a bespoke version of the Google Earth Engine Digitisation Tool (GEEDiT; Lea (2018)) within Google Earth Engine's (GEE) API (Gorelick et al., 2017). This GEEDiT-TermPicks version builds substantially on the original GEEDiT, with improvements made to both the digitisation interface, metadata options, sensor availability, and image accessibility. A user guide is provided as an appendix to this paper. A major advantage of GEEDiT-TermPicks over traditional repository download and visualization approaches is that it accesses the archive of Landsat, Sentinel-1 and Sentinel-2 and

ASTER images on the Google Cloud servers within a standard web browser. It therefore allows for much faster access to imagery compared to the alternative of downloading, extracting, and processing each individual image. This is combined with an interface for easy digitisation of margins that now uses GEE's DrawingTools functions to improve both speed and flexibility of digitisation for users.

To ensure that future data generated using this tool will be consistent with our dataset, the GEEDiT-TermPicks interface

visualises the TermPicks ID locations, allowing the user to easily identify the glaciers present and access relevant imagery. Once a glacier is chosen, GEEDiT-TermPicks provides rapid access to all available satellite images of that glacier, which can be pre-filtered by date and satellite. If the image is clear, the termini can be extracted by simply clicking on the screen along the glacier margin. Images with glacier termini that are low in quality can be compared with previous or subsequent images that are nearby in date to help better determine the location of the terminus for a specific date/time. If this is done, it will

automatically be flagged in the image metadata, though this (and other) image quality flag options can be manually selected, including options to provide a written note as to why the image is inadequate. Data exported from GEEDiT-TermPicks will therefore include as standard all metadata required for easy inclusion into future TermPicks data releases.

Finally, we recommend a minimum of 11 vertices per km of trace for quality that is consistent with this database. We also recommend tracing across the entire width of the glacier terminus as previous studies have shown that information about mass

loss processes can be obtained from studying the map-view change in trace morphology at high levels of detail (Fried et al., 2018; Chauché et al., 2014).

# 5  Conclusions

We present a new compilation of outlet glacier terminus traces for the GrIS spanning a time period from 1916 to 2020 obtained through manual tracing of the ice-ocean boundary. Data were cleaned, reformatted, assigned to image image IDs, and quality

controlled for use in machine learning algorithms that will enable semi-automated terminus tracing. Termini are provided in the same format and with similar metadata to ongoing machine learning-based terminus tracing. We have combined TermPicks data with that from CALFIN (Cheng et al., 2020) in our data repository. We find errors in TermPicks on the order of ∼100 m, similar to machine-identified termini. We find biases in terms of data coverage with well-studied glaciers with high coverage of terminus trace data, and other glaciers devoid of consistent coverage, showcasing the need for further manual and machine-

learning efforts to provide terminus data. We provide tools for future tracing efforts and include software to enable the use of these data for the broader scientific community.

*Code availability.*  This work was performed using freely-available software, primarily Google Earth, Google Earth Explorer, Python, and QGIS. Code to generate a text file that includes the Digital Object Identifier of citations for users is available on the GitHub site (https://github.com/sgoliber/TermPicks). GEEDiT TermPicks can be accessed through Google Earth Engine Code Editor

( https://github.com/jmlea16/GEEDiT-TermPicks)

*Data availability.*  Terminus trace data will be made available at NSIDC, a NASA DACC. Until the data submission is approved, data are currently available on Zenodo (https://doi.org/10.5281/zenodo.6557981). A shapefile of combined CALFIN and TermPicks data is included in this repository. We also provide terminus retreat data, the TermPicks ID shapefile, and kmz file that can be viewed in Google Earth and provides a quick look of temporal coverage (compared to imagery availability) for all glaciers.

*Author contributions.*  SG collected, cleaned, and formatted the data and metadata and performed most of the analyses. TB performed some data analysis and synthesis. SG, TB, and GC wrote the manuscript. JL created GEEDiT TermPicks. DC provided expertise on machine learning. All authors contributed to the editing and refining of the manuscript and data contribution.

*Competing interests.*  No competing interests are present

*Acknowledgements.*  We acknowledge support from a NASA Earth and Space Sciences fellowship to Goliber (18-EARTH18F-323) and
terminus tracers everywhere. At the University of Texas this includes: Lucero Casteneda, Gene Hsu, David Peters, Andrea Jimenez, Mason Fried. NJK was supported by the Programme for Monitoring of the Greenland ice sheet (PROMICE). MW was supported by an appointment to the NASA Postdoctoral Program at the Jet Propulsion Laboratory, California Institute of Technology, administered by the Universities Space Research Association under contract with NASA. JML is supported by a UKRI Future Leaders Fellowship (Grant No. MR/S017232/1). DF acknowledges support for this study through the EPSRC and ESRC Centre for Doctoral Training on Quantification and Management
of Risk and Uncertainty in Complex Systems Environments Grant No. EP/L015927/1. TM is funded by The Leverhulme Trust Research

Leadership scheme F/00391/J and the UK NERC NE/G010366/1. Landsat images used in figures were downloaded from USGS Earth Explorer https://earthexplorer.usgs.gov/.

# Appendix

## A  Google earth package

The TermPicks.kmz is a Google Earth KML (Keyhole Markup language) file and supporting images with the Landsat coverage for each glacier. Users can select their glacier of interest either in the side menu, or by navigating Google Earth to the glacier. Clicking on the glacier ID gives an image pop up that is the same format as Figure 11. This can be used to get a overview of what the data coverage is for each glacier; however, it does not include other data sources, such as Sentinel data. This file can be found in the TermPicks GitHub repository.

## B  TermPicks reference polygons

The TermPicks polylines need to be converted into label polygons for the deep learning usage. The label polygon contains the glacier terminus, fjord boundary, and outer boundary that ensures the polygon covers the corresponding remote sensing image. To convert into polygons, we first prepare a reference label polygon for each glacier. The terminus position of the reference polygon should be at the furthest retreat position so that the fjord boundary is exposed to the most extent. Then, for

any given terminus in TermPicks, we use it to replace the terminus part of the corresponding reference polygon to generate label polygons. A file containing reference polygons for the current TermPicks dataset is currently available on Zenodo (https://doi.org/10.5281/zenodo.6557981).

## C  GEEDiT-TermPicks

GEEDiT-TermPicks is written within Google Earth Engine's (GEE) API (Gorelick et al., 2017). This bespoke version of

GEEDiT (Lea, 2018) provides much the same functionality as the original, though represents a significant re-writing of its structure to allow for several improvements and TermPicks specific requirements, including:

1. Changing the digitisation interface so it operates using the Google Earth Engine DrawingTools functions (Google, 2021, link: https://developers.google.com/earth-engine/tutorials/community/drawing-tools, last accessed 5/21/2021). This allows even more rapid digitisation due to data being temporarily stored 'client side', rather than in the original tool where

vertices were submitted 'server side' for subsequent visualization (see Google, 2021, link: https://developers.google.com/earth-engine/guides/client_server for more information, last accessed 5/21/2021).

2. User skipping of images by date as well as image number

3. Inclusion of ASTER L1T Radiance image archive (https://lpdaac.usgs.gov/products/ast_l1tv003/, last accessed 5/21/2021)

4. Easy user access to imagery of glaciers that make up the TermPicks database and their respective locations.

5. Automatic appending of glacier, imagery and digitisation metadata, allowing any future versions of TermPicks to be easily and quickly generated.

6. Compulsory fields for user names and email addresses that are appended in metadata to ensure that those who digitize the data are properly acknowledged if and when they are subsequently shared/published, and (where necessary) to enable user inter-comparisons.

## D  GEEDiT-TermPicks walkthrough

Link to GEEDiT-TermPicks: https://github.com/jmlea16/GEEDiT-TermPicks

Step 1:

1. Define date range, months of interest, maximum image cloud cover limit, and satellites to visualise imagery from. Note that maximum image cloud cover limit uses the metadata values indicating cloud cover across the entire image that are provided with Landsat, Sentinel 2, and ASTER imagery. See Figure A1 for overview of menu screen.

2. Zoom to the glacier of interest and click on its blue dot.

Step 2:

1. The tool will automatically zoom to the selected glacier, and the blue dot will turn red. Imagery in the background is the standard Google Earth base imagery. If you have selected the incorrect glacier, click the 'Go Back' button and this will return you to the previous screen. See Figure A2.

2. Enter your name and email address in the boxes provided, and click the 'Go to images' button to continue. These are compulsory fields to ensure that data can be appropriately acknowledged where they are shared/published.

Step 3:

1. Imagery for the selected glacier, satellite and date is displayed. Zoom to the desired level to allow accurate digitisation of the terminus, and click on the screen to start digitising, and double click to end. It is possible for multiple lines to be digitized per image, though if users are seeking consistency with the TermPicks dataset this should be avoided. See Figure A3.

2. Four panels are included on this screen, including:

    (a) Panel for adding/removing extra images for comparison, and modifying margins that have been digitized. Clicking 'Remove added images' will remove any images that have been added by the user, leaving only the original satellite image on the screen. The 'Edit' button can be used where a line has been finished, but needs to be subsequently

modified or deleted. To do this, click the 'Edit' button and then click on the line that needs to be modified. This will allow its vertices to be moved, while the line can be deleted by pressing the delete or backspace key while the line is selected. To switch back to drawing mode, press the 'Draw new line' button. This will allow a line to be digitized by clicking on the screen as before. See Figure A4.

(b) Panel for assigning quality flags. Each of these flags can be manually assigned by the user as appropriate, though for SLC-off Landsat 7 imagery, and where the user uses panel 1 to compare to other imagery, the relevant check boxes will automatically selected. If digitized margins fall in areas of SLC-off Landsat 7 images, or the user has added an extra image for comparison in error, these flags can be manually deselected. Values of all flags, and any text notes are automatically appended to margins as metadata when they are exported. See Figure A5.

(c) Panel displaying glacier name, TermPicks ID and satellite that collected the displayed image. Text boxes display the date of the displayed image in YYYY-MM-DD format, and the image number of the total available of the glacier. Users can also skip to different images, by date or image number. Where users choose to enter dates, they must be given in YYYY-MM-DD format, and the image shown will be the image that is the closest available in time to the entered date. If a user defined image number falls outwith the range of valid values the map will be cleared and a panel requesting the user to enter a valid number will appear. Once a date/image number has been entered, the user can skip to that image by pressing the enter key. See Figure A6.

(d) Panel that allows the user to skip to the next/previous image number, or export the entire set of digitized margins. By pressing any of these buttons, the user will log the digitized margins for export. Once any of these buttons have been pressed, subsequent modification of the data via the geometry imports bar will result in duplicate margins in the exported dataset. Pressing 'Export' will set up an export task that can be accessed through the 'Task Manager' tab in the top right of the screen (Google, 2021, https://developers.google.com/earth-engine/guides/playground, last accessed: 5/21/2021). To avoid the possibility of data loss through failure of internet connection and/or browser crashes, it is recommended that users regularly export their data. See Figure A7.

(e) By hovering the cursor over the Geometry Imports panel, users can view all previous termini that have been digitized. The name of each geometry is given in the format t_YYYY_MM_DD_HHmm, where the date and time are derived from the image. Margins from previous images are visualised in blue, and those for the current image in black by default. Note that any modifications to previous margins (*i.e.* blue lines) will not be logged. See Figure A8.

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

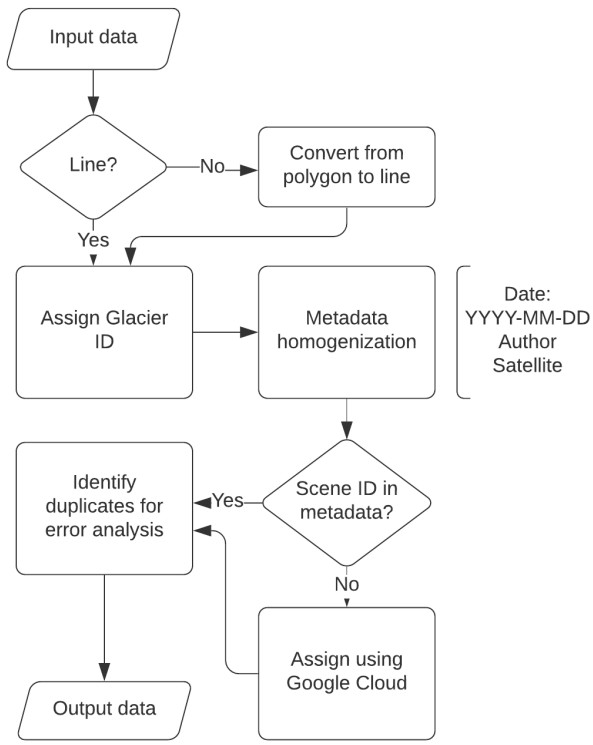

**Figure 1.** Flow chart showing processing pipeline for producing consistent terminus trace training data.

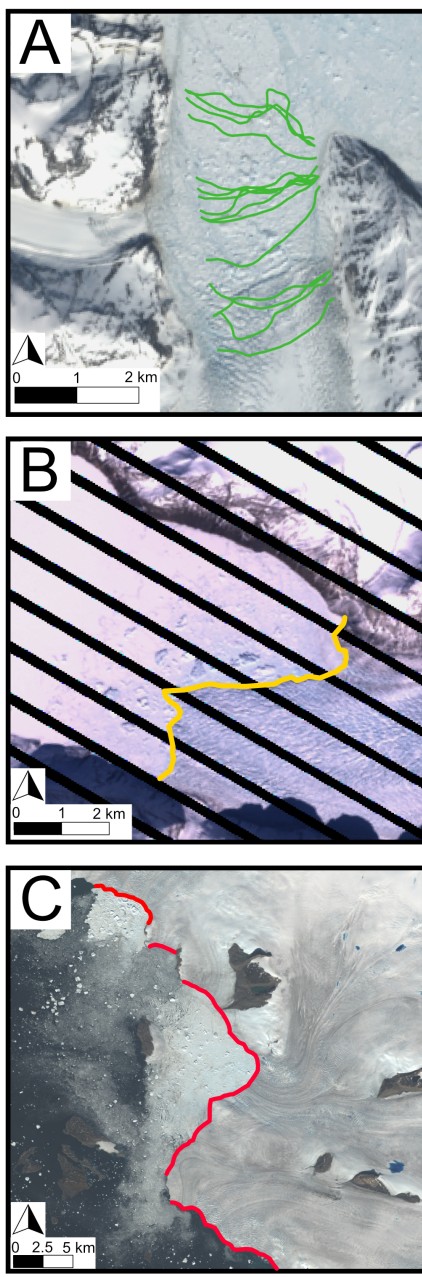

**Figure 2.** Common issues addressed in data cleaning and labeling. a) Box method glacier traces are contained within a box that is smaller than the full terminus width at Glacier 224 b) Landsat 7 ETM+ Scan Line Corrector-off image line artifacts at Glacier 291 and c) A single shapefile containing several different glaciers (IDs 27-30) that need to be split manually into separate glaciers to be consistent with the ID scheme. Additionally, all 3 images show varied levels of obstruction of the terminus in the fjord due to ice mélange. Landsat-7 and Landsat-8 images courtesy of the U.S. Geological Survey.

| GlacierID | Date | Year | Month | Day | DecDate | QualFlag | Satellite | ImageID | Author | Center_X | Center_Y |
|---|---|---|---|---|---|---|---|---|---|---|---|
| 3 | 2013-07-22 | 2013 | 7 | 22 | 2013.553425 | 05 | Landsat | LC08_L1TP_014009_20150911_20170404_01_T1 | Catania | -52.574288 | 72.042421 |
| 278 | 2018-02-28 | 2018 | 2 | 28 | 2018.158904 | 01, 04 | Sentinel 1 | SEN1_NSIDC_0723_V2_20180226_20180303 | Black_Taryn | -49.677742 | 69.203580 |
| 291 | 1978-07-03 | 1978 | 7 | 3 | 1978.501370 | 00 | Aerial Photo | 874D0164 | Korsgaard | -51.406852 | 71.461398 |

**Figure 3.** Example metadata for the TermPicks data set. Each column corresponds to the description in Section 2.4 (Metadata Creation).

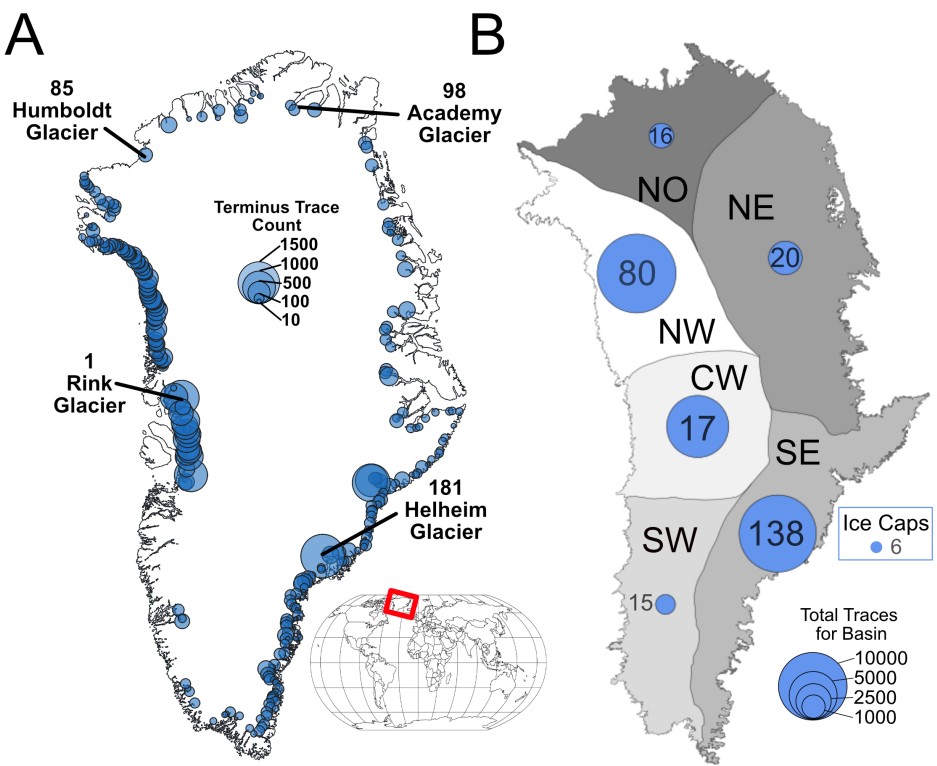

**Figure 4.** A) Terminus trace count for glaciers in Greenland. Each circle is centered on a location of a glacier in the TermPicks ID file. The size of the circle reflects the total number of terminus traces available for that glacier. B) The same data organized by drainage basin. Circle size reflects the total number of traces for that basin. The numbers inside or adjacent to the circle represent the number of individual glaciers in each basin with terminus traces. Each basin is defined by the ESA/NASA ice sheet mass balance inter-comparison exercise 2016 (IMBIE; Shepherd et al.) which includes basins from Rignot and Mouginot (2012) and Rignot et al. (2011). They are labeled by their geographic location. Region labels are NO = North, NE = North East, SE = South East, SW = South West, CW = Center West, NW = North West.

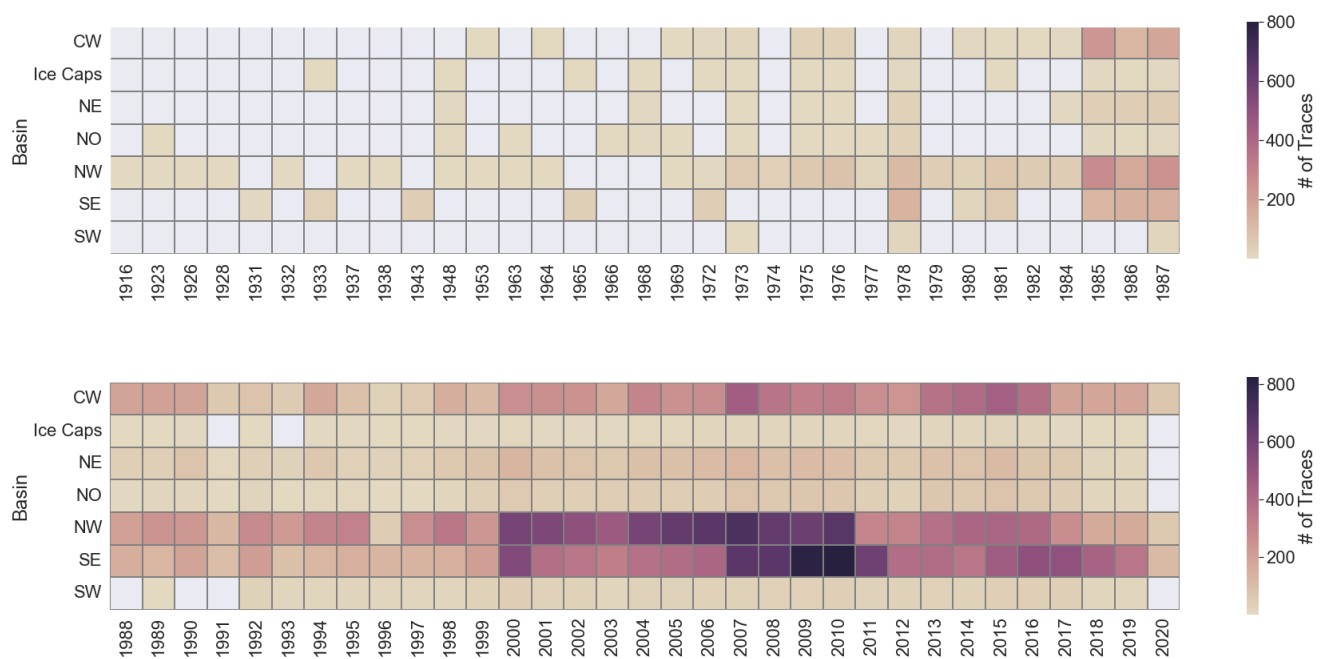

**Figure 5.** Heatmap of glacier traces in each regional basin from ESA/NASA ice sheet mass balance inter-comparison exercise 2016 (IMBIE; Shepherd et al.) in this study. Total number of traces per region can be found in Figure 4. The x-axis is year and the y-axis is the Basin ID. The color corresponds to the number of traces for that basin's glacier per year. 0 traces are grey.

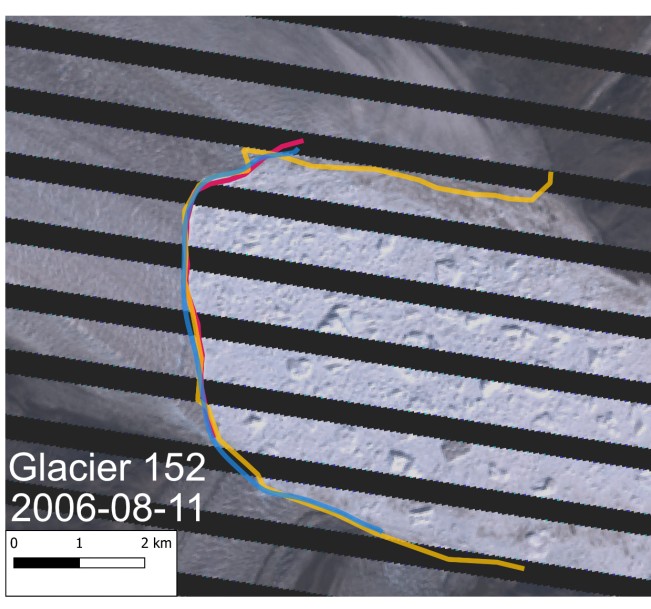

| | Retreat compared to 1978 position | |
|---|---|---|
| Author | Interpolation | Centerline |
| Bunce | 6.1 km | 6.5 km |
| Cheng | 5.9 km | 6.5 km |
| ESA | 4.6 km | 6.5 km |

**Figure 6.** Terminus positions for Glacier 152 (Kangerlussuaq Gletsjer) from 2006-08-11 for 3 authors. Bunce (pink) and Cheng (blue) traces end before the northern fjord wall while the ESA (yellow) trace ends at the northern wall. The table shows each calculated retreat amount since the 1978 position using the Interpolation method and the Centerline method. Landsat-7 image courtesy of the U.S. Geological Survey. Base image has reduced saturation to increase contrast with traces.

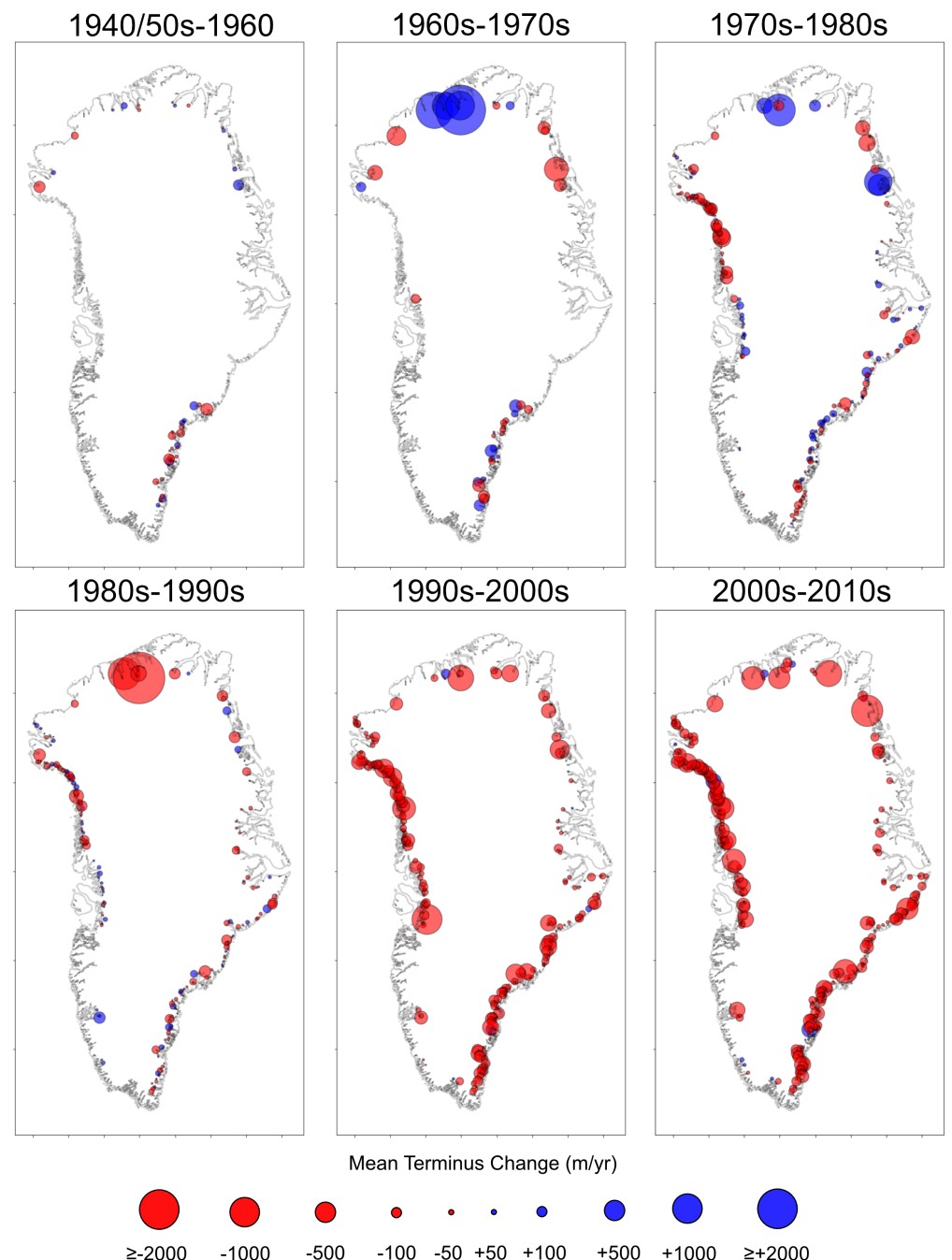

**Figure 7.** Decadal retreat patterns for available TermPicks data using the Centerline method. For each panel, the entire decade of traces were averaged to produce an average position for that decade. The 1940/1950s are an average over both decades as there are fewer traces available in the 1950s. Then the average position for the decade is differenced from average position of the previous decade. The size correlates to magnitude of terminus change, while red (negative) indicates retreat and blue (positive) indicates advance.

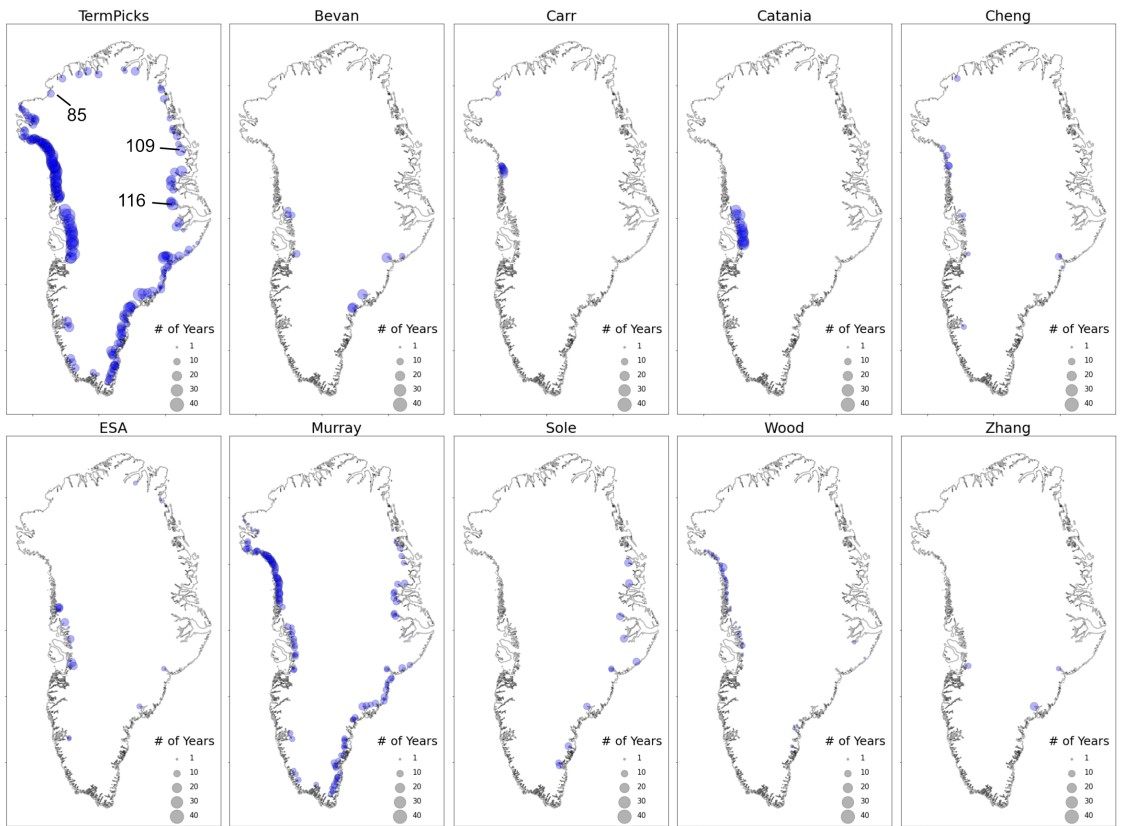

**Figure 8.** Locations of glaciers that include terminus delineations for at least three unique months, which is the minimum number of traces required to resolve seasonality, for the entire TermPicks data set and a subset of authors. The size of the blue circle indicates how many years there are enough traces to resolve seasonality, ranging between a single year to up to 40 years.

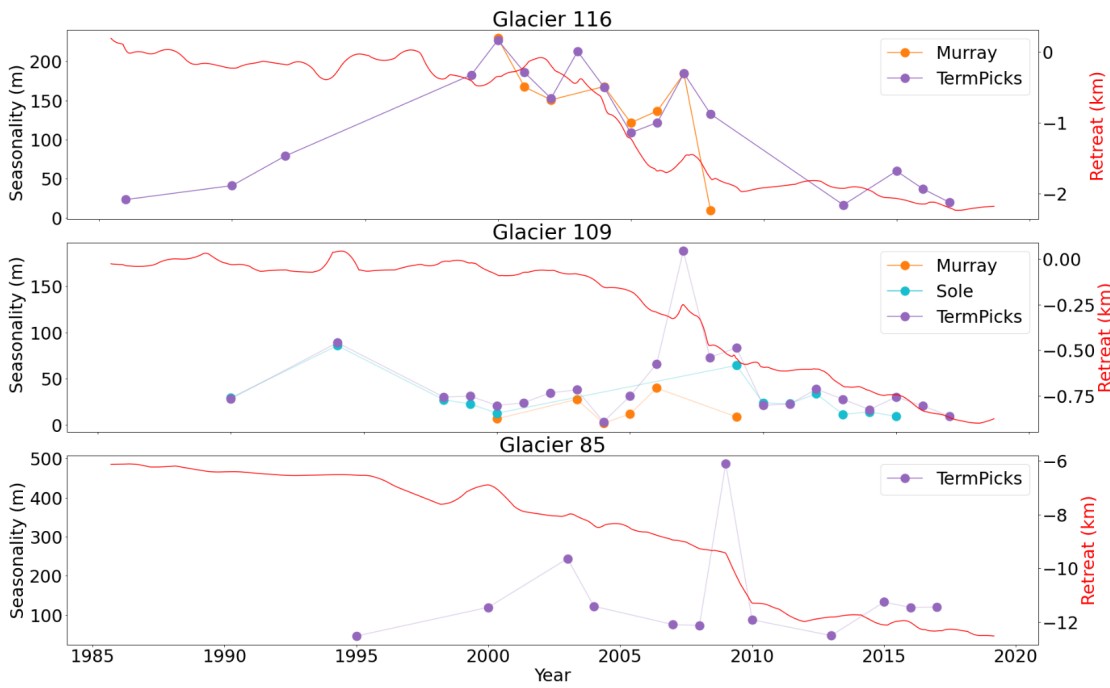

**Figure 9.** Example seasonality plots for three glaciers, F. Graae Gletscher(116), Heinkel Gletsjer (109), and Humboldt Gletsjer (85). The location of each of these glaciers is noted in Figure 8. Each color corresponds to either the entire TermPicks data set (purple) or an individual author. Glacier 85 has no individual author data set that can resolve the seasonality.

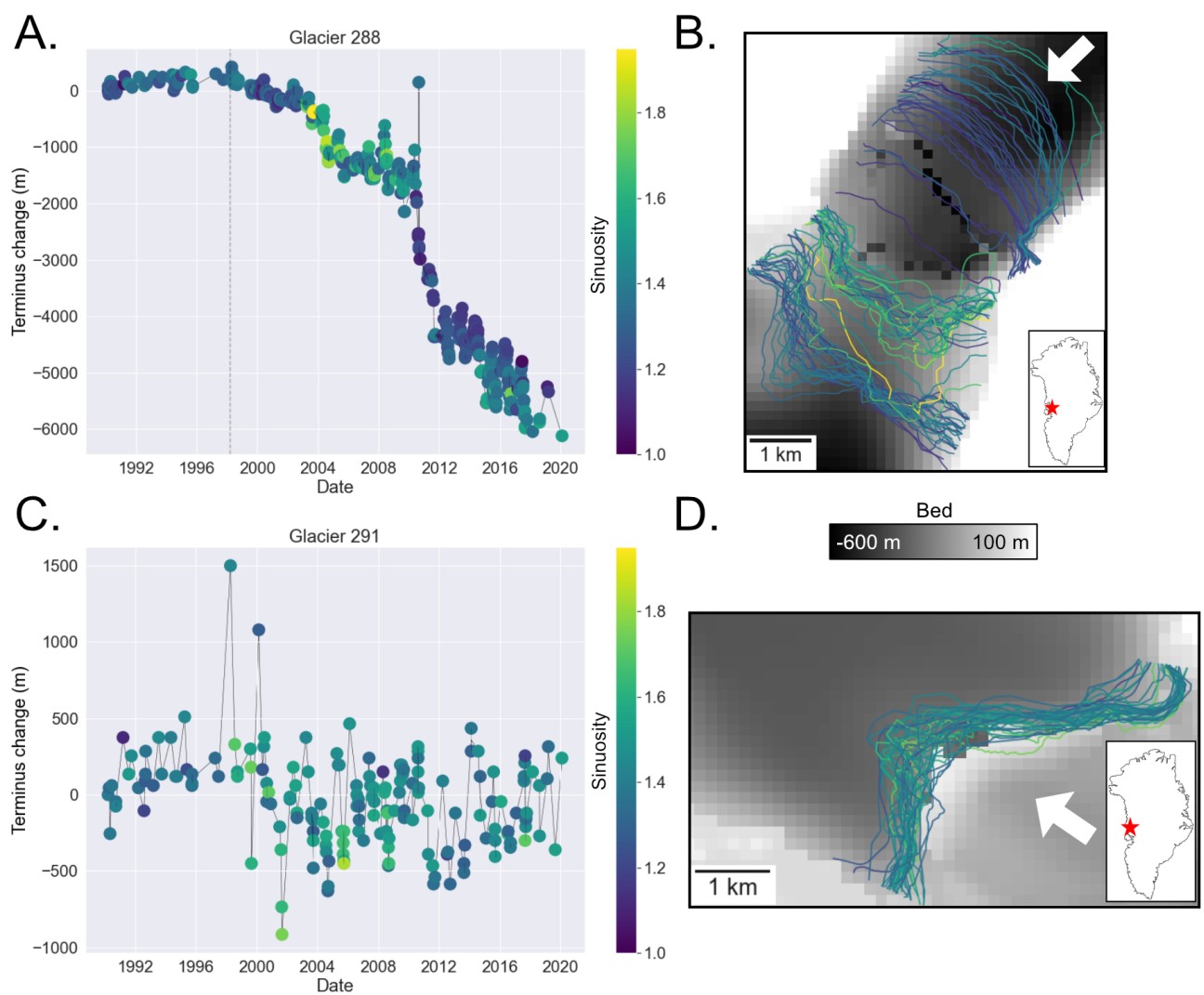

**Figure 10.** A: Terminus change between 1990-2020 colored by sinuosity for Glacier 288 (Sermeq Silarleq). The dashed grey line is the start of progressive retreat as defined in Catania et al. (2018). B: Corresponding map-view terminus traces For Glacier 288 with every 5th trace colored by sinuosity. C: Terminus change between 1990-2020 colored by sinuosity for Glacier 291. D: Corresponding map-view terminus traces for Glacier 291 (Kangerdlugssup Sermerssua) with every 5th trace colored by sinuosity. The base map in B and D is the bed from BedMachine (Morlighem et al., 2017). The black pixels in B are errors, however they do not impact the overall interpretation of the bed. The bed scalebar applies to both B and D. The white arrows indicate glacier flow direction. The red star on the inset map is the location of the glacier on the Greenland Ice Sheet.

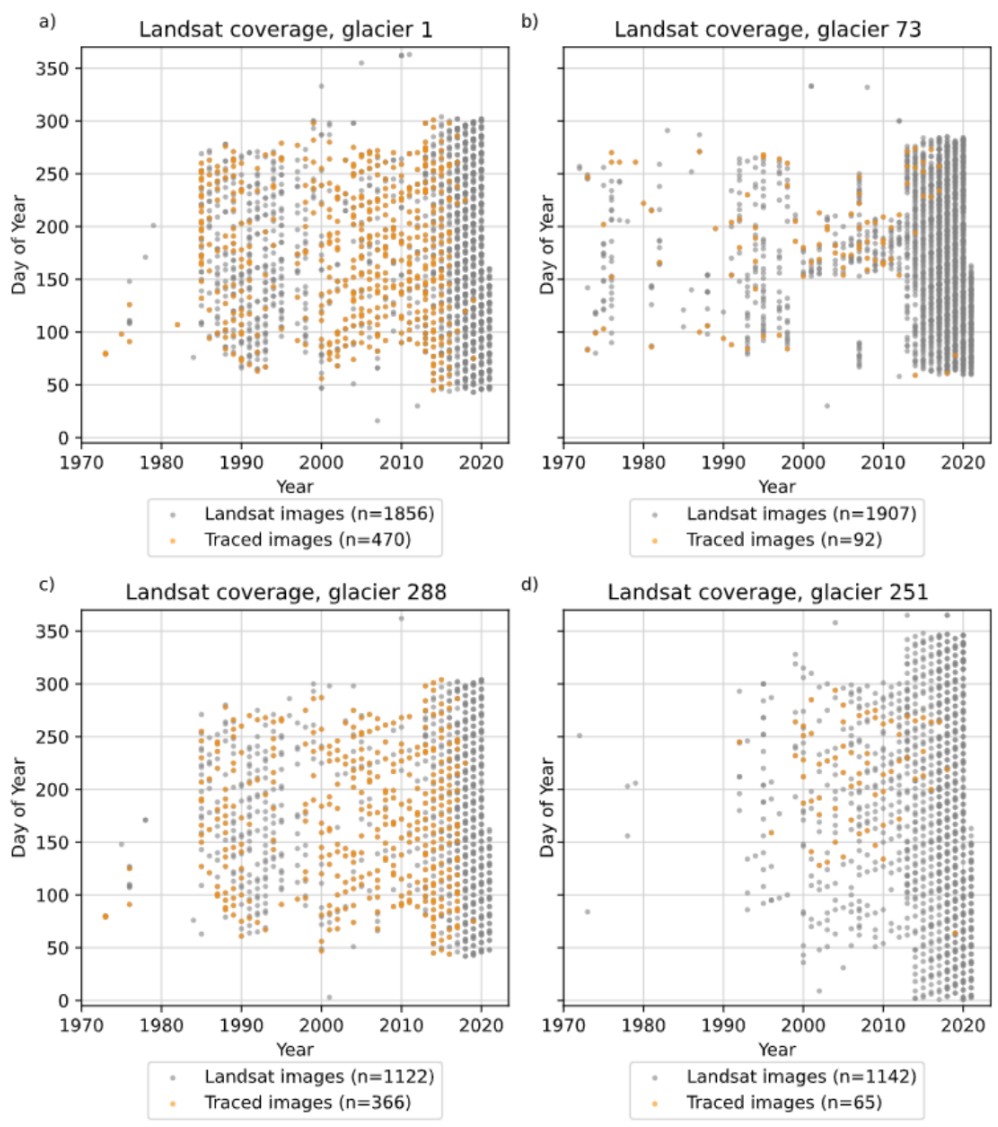

**Figure 11.** Examples of Landsat image availability (gray) versus termini traced (orange) for (a) Kangilliup Sermia (Rink Isbræ; 1), a relatively well-traced glacier, (b) Qeqertaarsuusarsuup Sermia (Tracy Gletsjer; 73), a glacier representative of the average number of total traces for this dataset, (c) Sermeq Silarleq (288), the glacier with the highest percentage of available Landsat images that have been traced in this dataset, and (d) an unnamed glacier (251), representative of the average percentage of available Landsat images that have been traced in this dataset.

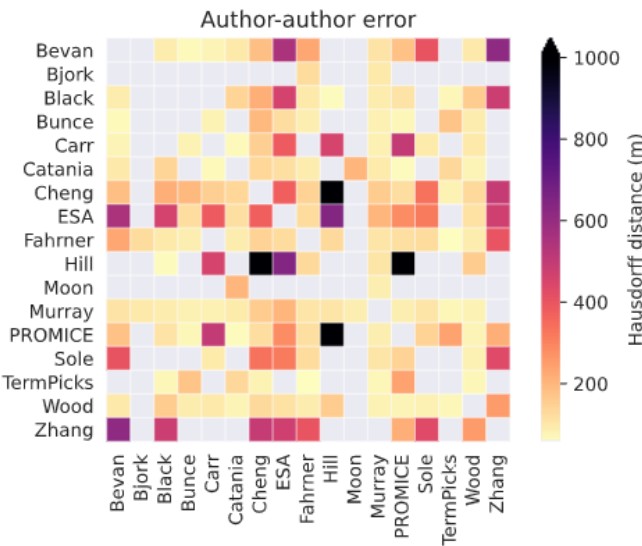

**Figure 12.** Median error between pairs of authors, for instances where those authors have duplicated a glacier trace on a given date. No color indicates two authors have no duplicated traces between them.

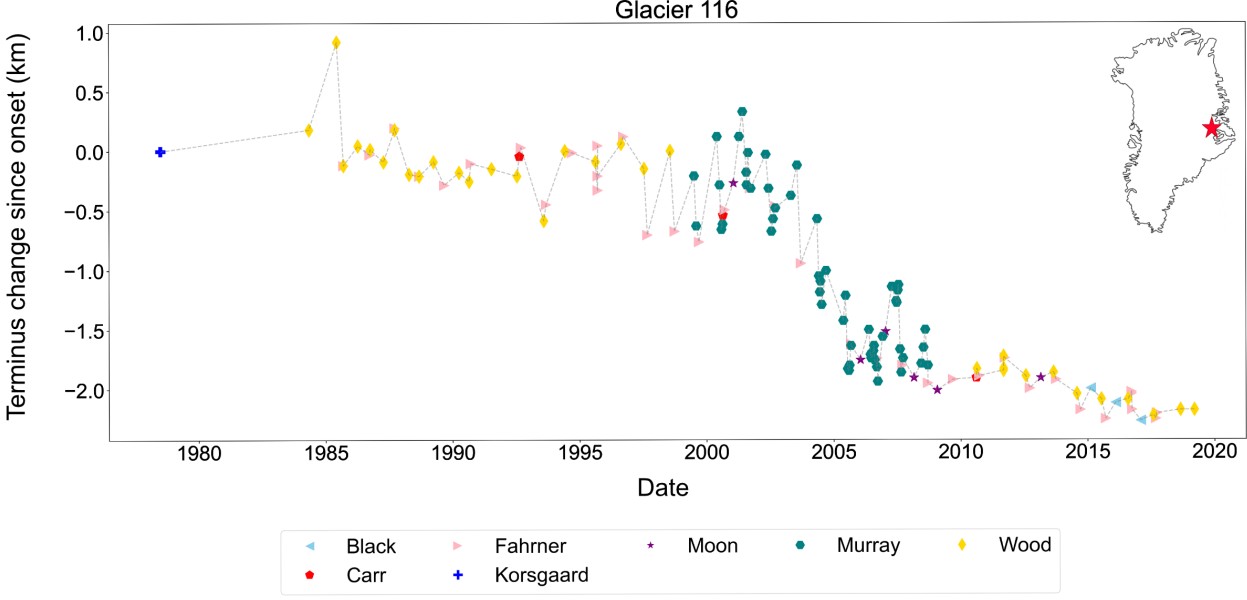

**Figure 13.** Example terminus change for Glacier 116 (F. Graae Gletscher). Color and symbol correspond to different authors for each pick.

| Published source | Spatial coverage | Date range | Resolution | Method | Author key |
|---|---|---|---|---|---|
| Andersen et al. (2019) | GrIS wide; n =47 | 1999-2018 | Annual | Full width | PROMICE |
| Bevan et al. (2012) | GrIS wide; n = 14 | 1985-2011 | Sub-annual | Full width | Bevan |
| Bevan et al. (2019) | Kangerlussuaq; n = 1 | 1985-2018 | Sub-annual | Full width | Bevan |
| Bjørk et al. (2012) | SE GrIS, n =132 | 1931-2010 | Decadal-sub-decadal | Full width | Bjork |
| Black and Joughin (2022) | NW GrIS; n = 87 | 1972-2021 | Annual | Box | Black |
| Brough et al. (2019) | Kangerlussuaq, n = 1 | 2013-2018 | Sub-annual | Box | Brough |
| Bunce et al. (2018) | NW and SE; n = 276 | 2000-2015 | Annual | Box | Bunce |
| Carr et al. (2013) | NW GrIS; n = 10 | 1976-2012 | Decadal to monthly | Box | Carr |
| Carr et al. (2017) | GrIS Wide; n = 273 | 1992-2010 | Decadal | Box | Carr |
| Carr et al. (2015) | Humboldt ; n = 1 | 1975-2012 | Decadal-sub-decadal | Full width | Carr |
| Catania et al. (2018) | CW GrIS; n = 15 | 1965-2018 | Sub-annual | Full width | Catania |
| Cheng et al. (2020) | GrIS wide; n = 65 | 1972-2019 | Sub-annual | Full width | Cheng |
| Cowton et al. (2018) | E GrIS; n = 10 | 1993–2012 | Sub-annual | Box | Sole |
| Fahrner et al. (2021) | GrIS wide; n = 224 | 1984–2017 | Annual | Full Width | Fahrner |
| Hill et al. (2017) | N GrIS; n = 21 | 1916-2015 | Annual | Box | Hill |
| Hill et al. (2018) | N GrIS; n = 18 | 1948-2015 | Annual | Box | Hill |
| Korsgaard (2021) | GrIS Wide; n = 452 | 1978–1987 | Annual | Full width | Korsgaard |
| Moon and Joughin (2008) | GrIS wide; n = 203 | 1992-2007 | Sub-decadal | Box | Moon |
| Murray et al. (2015a) | GrIS wide; n = 199 | 2000-2010 | Sub-annual | Full width | Murray |
| Raup et al. (2007) | GrIS wide; n = 28 | 1990-2016 | Sub-annual | Full width | ESA |
| TermPicks | E and W GrIS; n = 13 | 1985-2019 | Sub-annual | Full width | TermPicks |
| Wood et al. (2021) | GrIS wide, n = 226 | 1992-2017 | Annual | Full width | Wood |
| Zhang et al. (2019) | Helheim, Jakob., and Kanger.; n = 3 | 2009-2015 | Sub-annual | Full width | Zhang |

**Table 1.** Original sources for terminus traces for the TermPicks data set. Spatial coverage describes the number of glaciers and name/region(s) of the traces. Date range are the years covered by the data set. Resolution is the temporal resolution; Annual is approximately one trace per year, sub-annual is more than one trace per year, decadal is approximately one trace every ten years, sub-decadal is more than one trace every 10 years, but not each year. Method is the tracing method used by the author to digitize the terminus. The Author key is the label given to that data set in the TermPicks data set.

| Source name | Start date | End date | Spatial res. (m) | Temporal res. (days) | Sensor type |
|---|---|---|---|---|---|
| ASTER | 01-2000 | 11-2020 | 15-19 | 16 | Multispectral |
| Landsat 1 | 07-1972 | 01-1978 | 80 | 18 | Multispectral |
| Landsat 2 | 01-1975 | 08-1983 | 80 | 18 | Multispectral |
| Landsat 3 | 03-1978 | 09-1983 | 80 | 18 | Multispectral |
| Landsat 4 | 07-1982 | 12-1993 | 30 | 16 | Multispectral |
| Landsat 5 | 03-1984 | 01-2013 | 30 | 16 | Multispectral |
| Landsat 7 | 04-1999 | Ongoing | 30 | 16 | Multispectral |
| Landsat 8 | 02-2013 | Ongoing | 30 | 16 | Multispectral |
| Sentinel 1 | 04-2014 | Ongoing | 20 | 6-12 | SAR |
| Sentinel 2 | 06-2015 | Ongoing | 10 | 12 | Multispectral |
| SPOT-1 | 02-1986 | 12-1990 | 20 | 26 | Multispectral |
| Corona | 06-1959 | 05-1972 | 7.5 | Irregular | Photograph |
| 7th Thule Expedition Aerial Oblique Photos | 1933 | 1933 | | Single | Photograph |
| British Arctic Air Route Expedition (BAARE) | 1931 | 1931 | | Single | Photograph |
| Danish aerial photos | 1978 | 1987 | | Single | Photograph |
| US Navy/US Army Air Force | 1943 | 1943 | | Single | Photograph |
| ALOS-PALSAR | 01-2006 | 04-2011 | 10-20 | 14 | SAR |
| ENVISAT | 03-2002 | 04-2012 | 30 | 35 | SAR |
| ERS-1 | 07-1991 | 03-2000 | 30 | 3, 35, and 168 | SAR |
| ERS-2 | 04-1995 | 09-2011 | 30 | 3, 35, and 168 | SAR |
| JERS-1/ Fuyo-1 | 02-1992 | 10-1998 | 18 | 44 | SAR |
| TerraSAR-X | 01-2008 | 12-2020 | 40 | 11 | SAR |
| RADARSAT 1 | 11-1995 | 03-2013 | 100 | 11 | SAR |

**Table 2.** Image sources used in this compilation of manually-traced glacier terminus trace dataset.

| Flag code | Issue |
|---|---|
| X = 0 | Manually-digitized trace |
| X = 1 | Machine-generated trace |
| X0 | No issues |
| X1 | Trace uncertainty due to environment or image issues (clouds, shadows, missing data, etc.) |
| X2 | Supplemented trace |
| X3 | Landsat 7 SLC off |
| X4 | Incomplete/Box Method |
| X5 | Automatically assigned Image ID |

**Table 3.** Flags assigned to output terminus trace data, created in conjunction with CALFIN (Cheng et al., 2020). All data in the TermPicks dataset has the prefix of X = 0.

| Author | Vertices per km | Mean median error (m) | Median median error (m) |
|---|---|---|---|
| Bevan | 2.5 | 227.5 | 145.8 |
| Bjørk | 14.2 | 113.6 | 113.6 |
| Black | 5.7 | 181.9 | 111.2 |
| Brough | N/A | N/A | N/A |
| Bunce | 14.1 | 109.0 | 88.3 |
| Carr | 7.1 | 201.0 | 98.0 |
| Catania | 18.3 | 112.7 | 100.9 |
| Cheng | 211.1 | 720.5 | 171.8 |
| ESA | 10.4 | 321.9 | 317.8 |
| Fahrner | 5.9 | 139.3 | 122.5 |
| Hill | 10.0 | 1458.8 | 309.1 |
| Korsgaard | 9.7 | N/A | N/A |
| Moon | 5.5 | 148.0 | 148.0 |
| Murray | 6.3 | 106.7 | 96.5 |
| PROMICE | 16.5 | 355.5 | 133.2 |
| Sole | 5.4 | 228.1 | 144.5 |
| TermPicks | 11.8 | 113.7 | 78.7 |
| Wood | 23.1 | 114.5 | 96.7 |
| Zhang | 55.7 | 421.8 | 452.0 |

**Table 4.** Mean vertices per kilometer of trace, and mean and median of the median errors of each author compared to other authors.

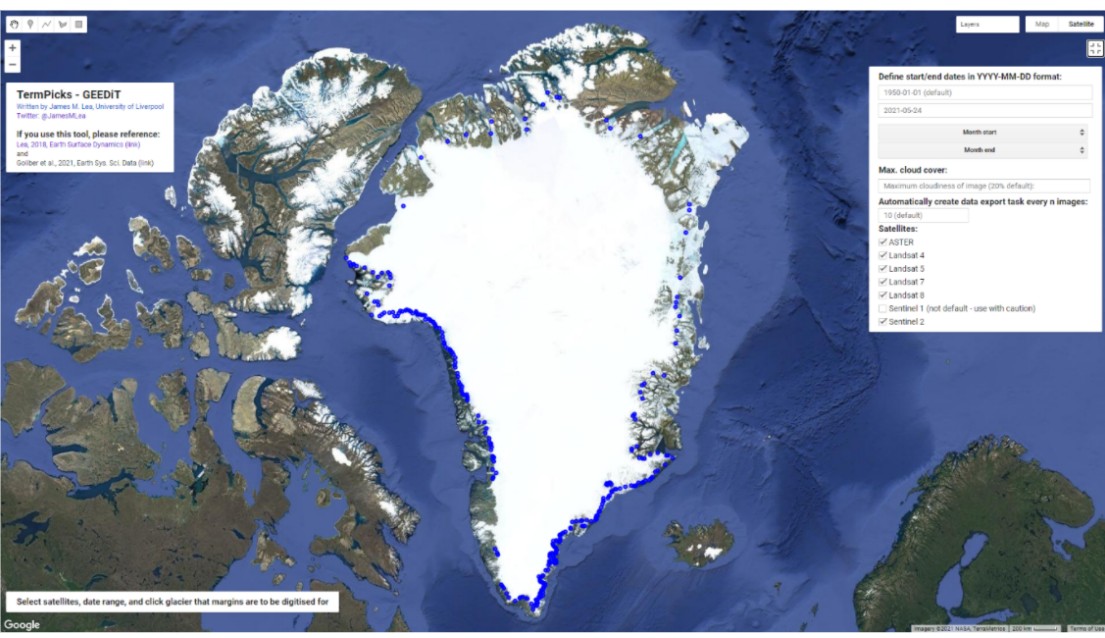

**Figure A1.** Step 1 of GEEDiT-TermPicks walkthrough. Overview of the menu screen (Screenshot from Google Earth Engine © Google Earth 2021).

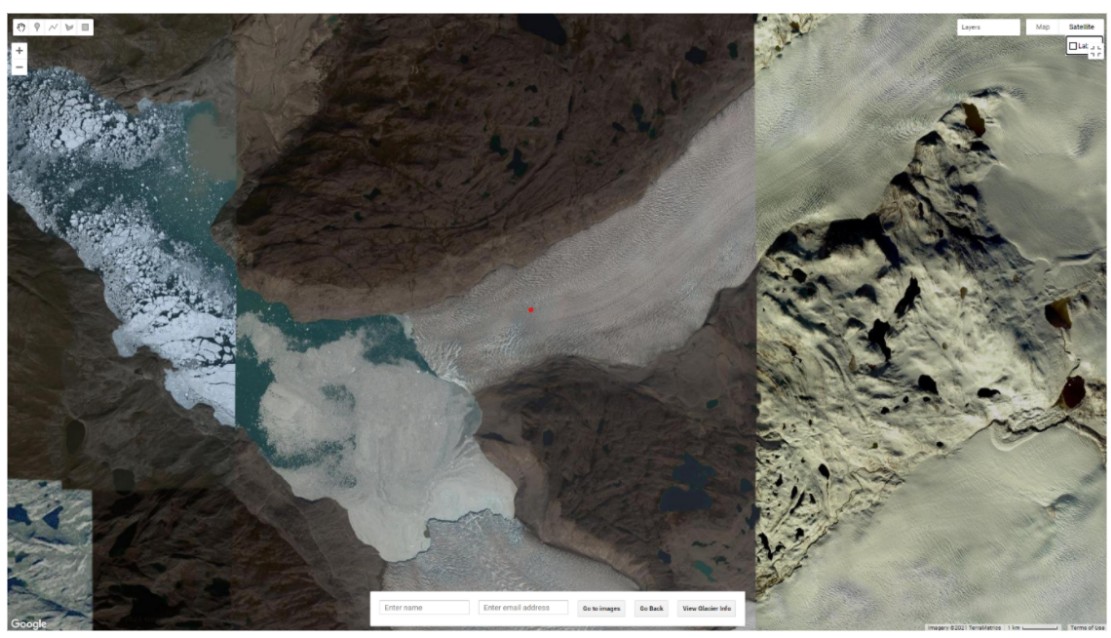

**Figure A2.** Step 2 of GEEDiT-TermPicks walkthrough. Zoom in of the individual glacier of interest menu (Screenshot from Google Earth Engine © Google Earth 2021).

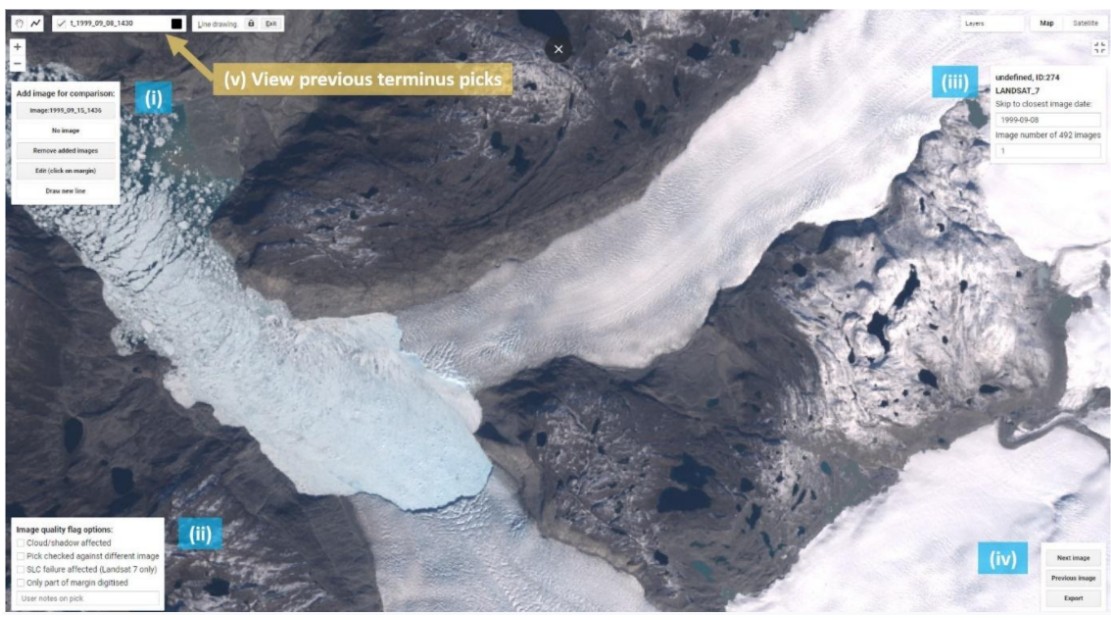

**Figure A3.** Step 3 of GEEDiT-TermPicks walkthrough. Zoom in of the individual glacier of interest menu and additional menus (Screenshot from Google Earth Engine © Google Earth 2021).

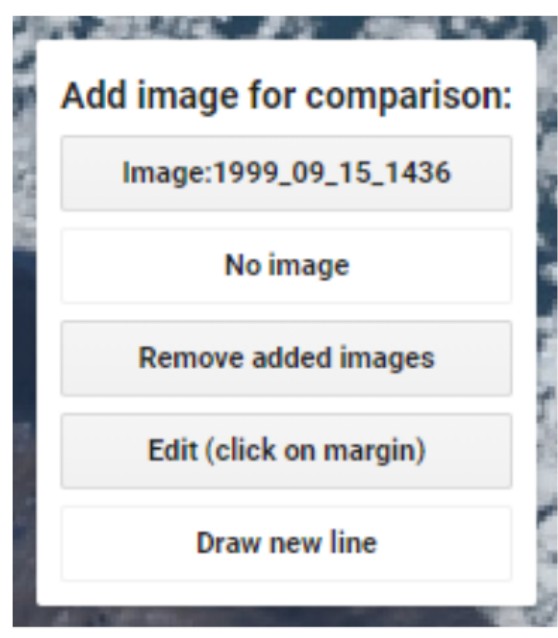

**Figure A4.** Step 3a of GEEDiT-TermPicks walkthrough. Panel for adding/removing extra images for comparison, and modifying margins that have been digitized (Screenshot from Google Earth Engine © Google Earth 2021).

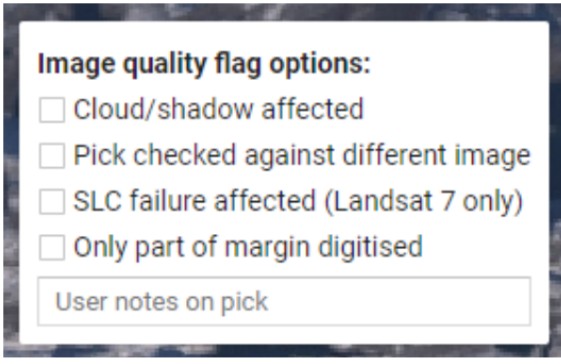

**Figure A5.** Step 3b of GEEDiT-TermPicks walkthrough. Panel for assigning quality flags (Screenshot from Google Earth Engine © Google Earth 2021).

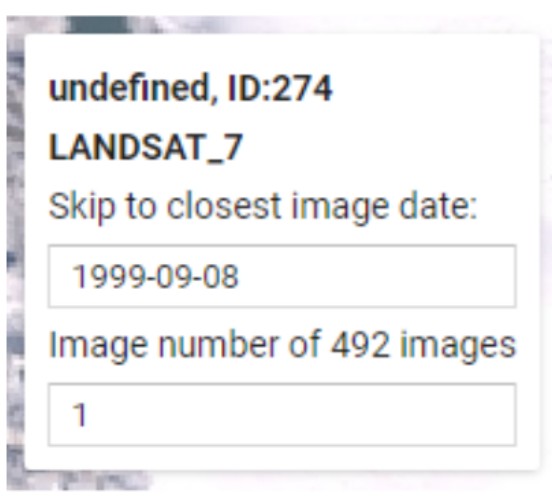

**Figure A6.** Step 3c of GEEDiT-TermPicks walkthrough. Panel displaying glacier name (Screenshot from Google Earth Engine © Google Earth 2021).

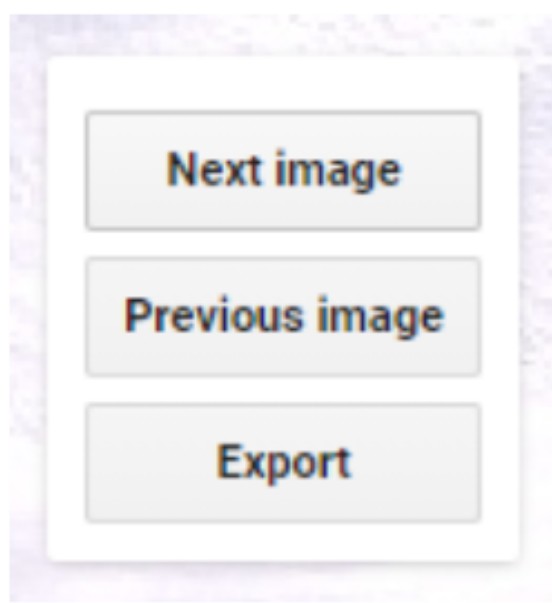

**Figure A7.** Step 3d of GEEDiT-TermPicks walkthrough. Panel that allows the user to skip to the next/previous image number, or export the entire set of digitized margins (Screenshot from Google Earth Engine © Google Earth 2021).

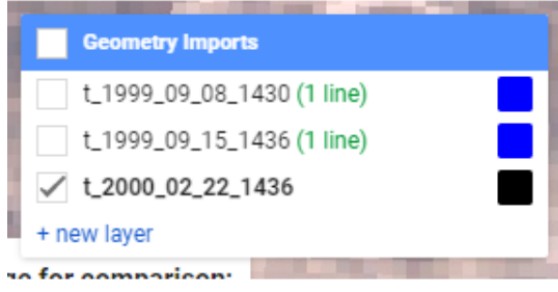

**Figure A8.** Step 3e of GEEDiT-TermPicks walkthrough. Panel of geometry imports to view previous termini (Screenshot from Google Earth Engine © Google Earth 2021).

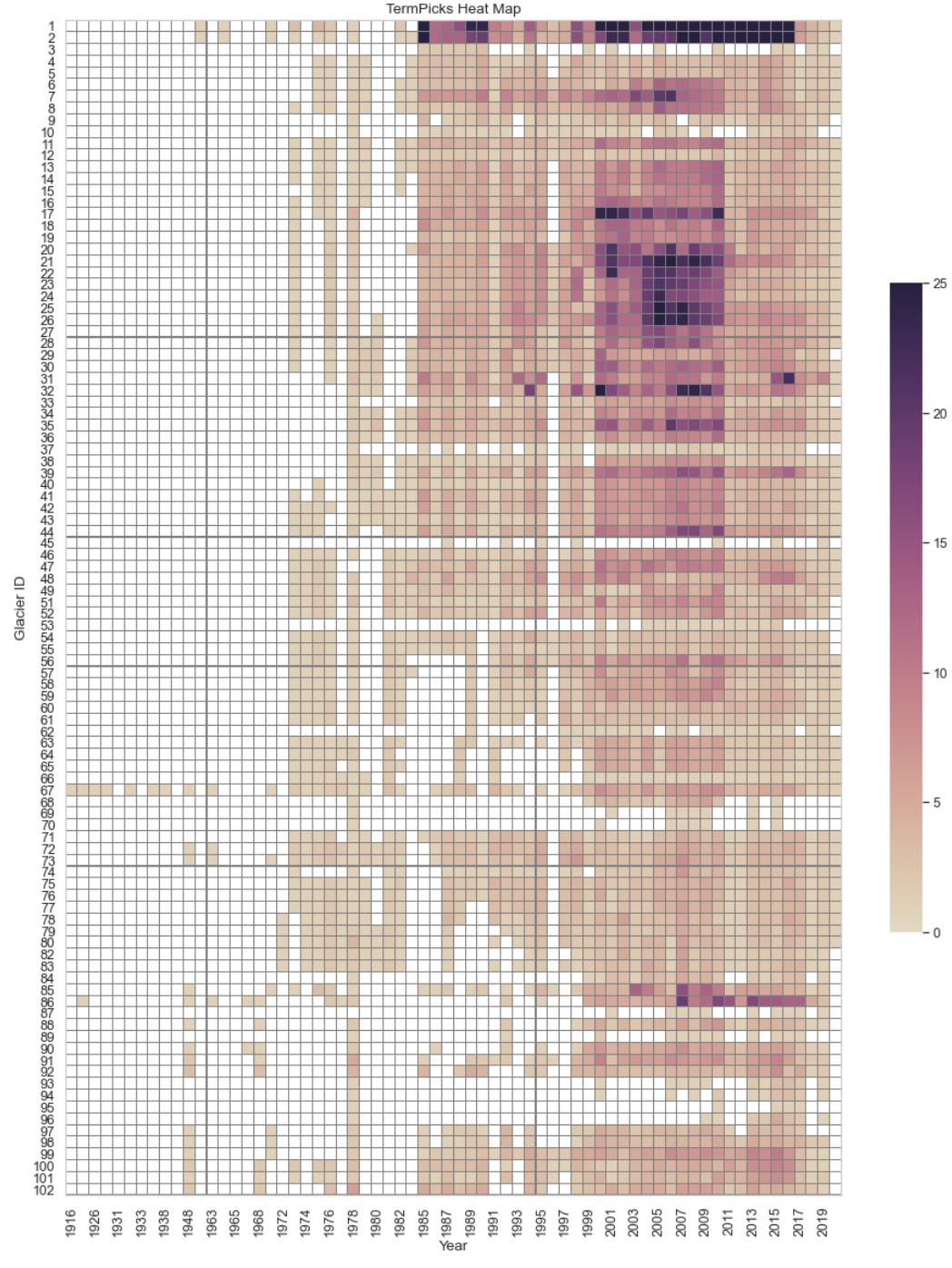

**Figure A9.** Heatmap of glacier traces for glaciers 1 to 102. The x-axis is year and the y-axis is the Glacier ID. The color corresponds to the number of traces for that basin's glacier per year, between 1 and >25. 0 traces are grey.

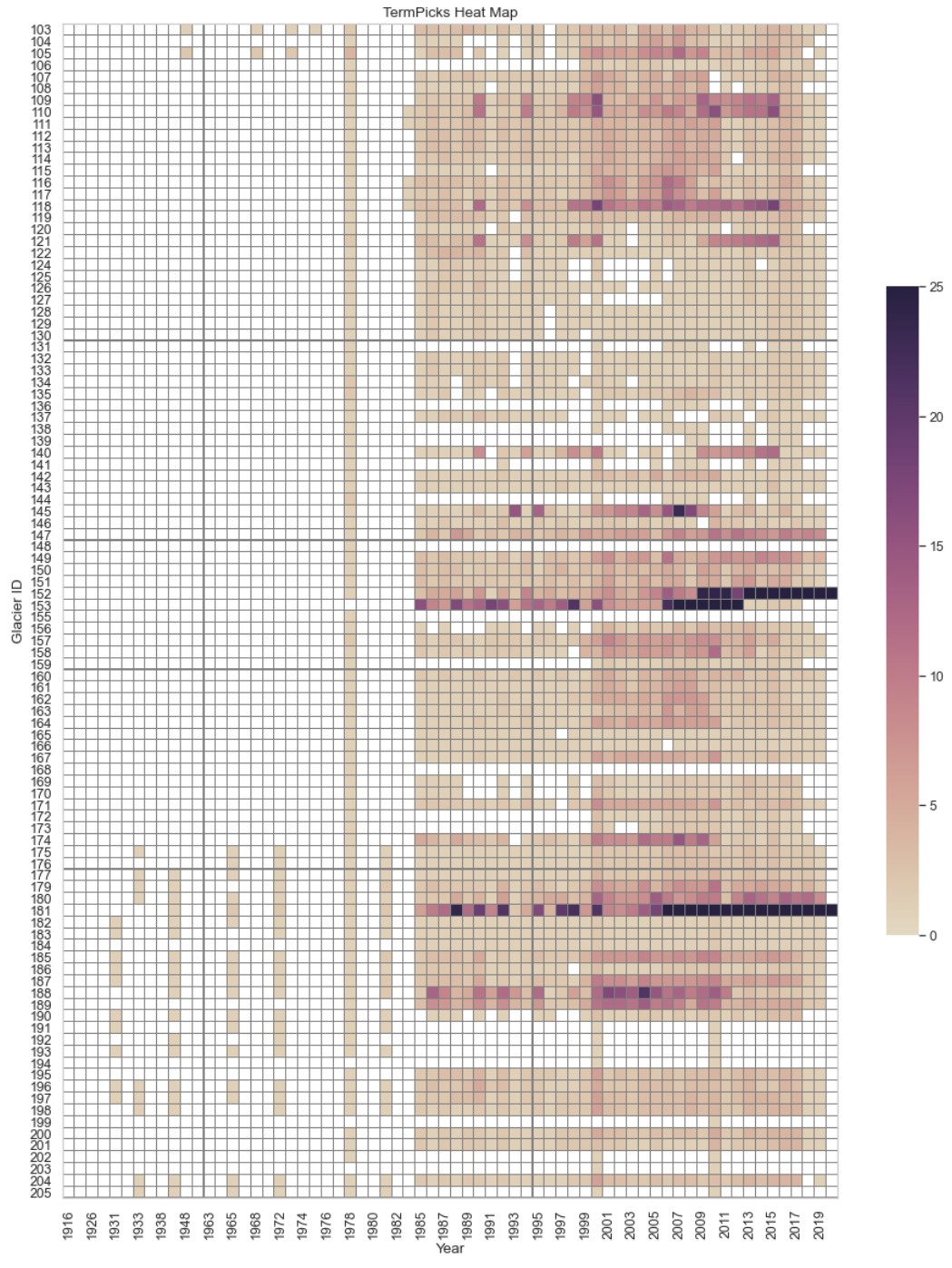

**Figure A10.** Heatmap of glacier traces for glaciers 103 to 205. The x-axis is year and the y-axis is the Glacier ID. The color corresponds to the number of traces for that basin's glacier per year, between 1 and >25. 0 traces are grey.

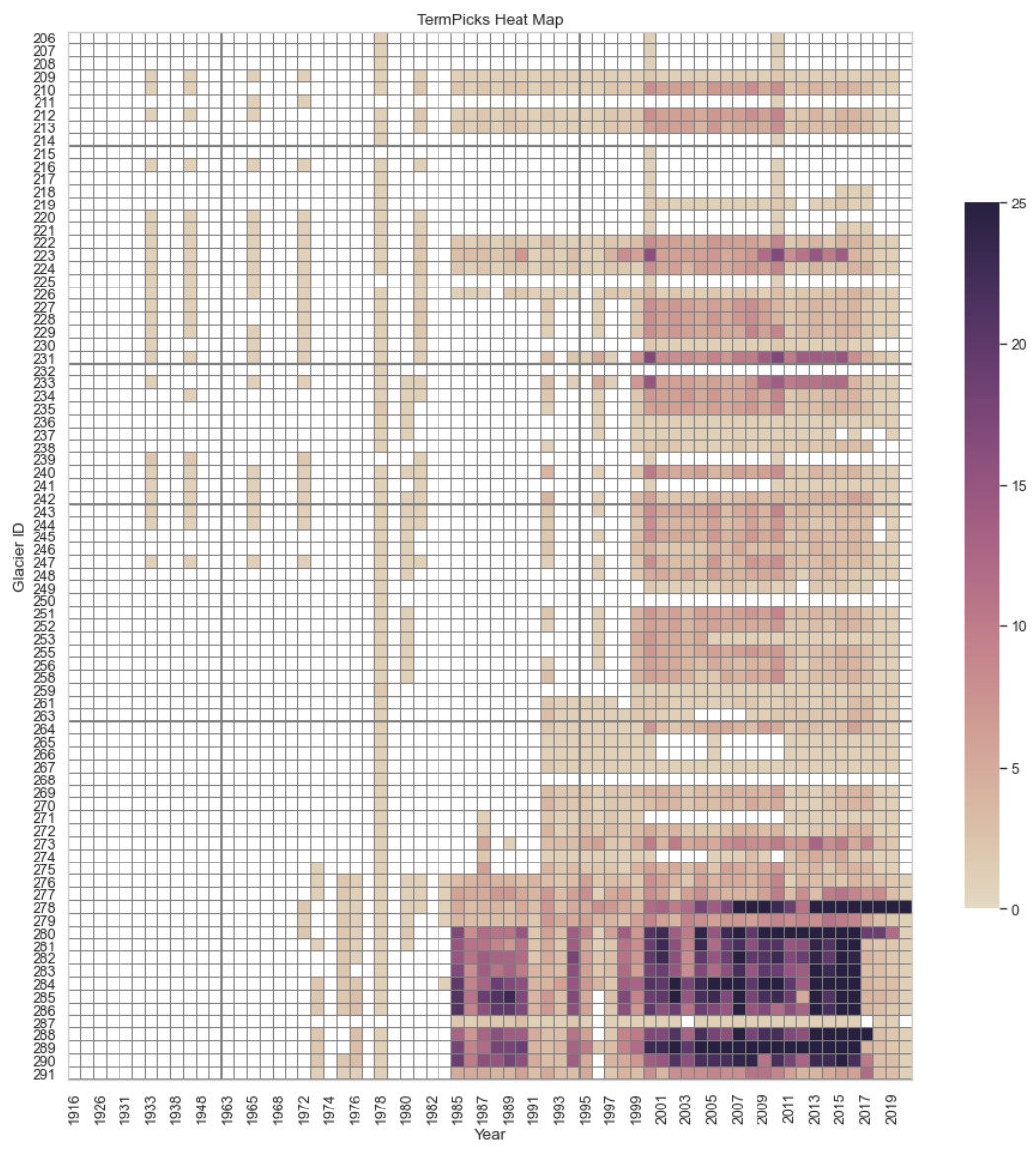

**Figure A11.** Heatmap of glacier traces for glaciers 206 to 291. The x-axis is year and the y-axis is the Glacier ID. The color corresponds to the number of traces for that basin's glacier per year, between 1 and >25. 0 traces are grey.