# Peer review of "TermPicks: A century of Greenland glacier terminus data for use in scientific and machine learning applications"

_The Cryosphere, 2021_

## Author Comment (AC1)

In this manuscript, the authors have described a dataset of manually digitized terminus positions for outlet glaciers of the Greenland ice sheet compiled from previously-published datasets, in order to provide a consistently-formatted training dataset for future machine learning applications. This is an excellent and timely undertaking that highlights the power of collaborative efforts.

On the whole, the manuscript does a good job describing the issues involved in combining "input" datasets from multiple authors, as well as describing the "ouptut" dataset, and even manages to show an example application of combining data sources. Accordingly, I only have a few minor comments/suggestions to make on the manuscript. The bulk of my comments/suggestions have to do with the description of the metadata - I think a Table with a few different example entries would help clarify this for a reader.

We appreciate the constructive and positive feedback on the manuscript. We address the comments bellow, but also included a new figure (10) of the metadata of three glaciers to better clarify the structure. As we addressed comments, the original line numbers of the text may have changed in the final manuscript. The changed text has been noted in the responses to individual comments. Our responses are in blue below each comment.

Comments to Address:

- l. 10: is this the mean (± standard deviation)?
  - Yes. This was changed to "The TermPicks data set includes 39,060 individual terminus traces for 278 glaciers with a mean of 136±190 and median of 93 of traces per glacier" to be more clear. The SD is higher than the mean due to the high variation of picks between certain glaciers.
- l. 52: check that months are removed from the reference dates
  - These have been checked and have removed from the text.
- l. 104: is the Howat reference here for the MODIS image?
  - The incorrect MEaSUREs image was being cited here. It has been changed to "MEaSUREs Greenland Ice Mapping Project (GIMP) 2000 Image Mosaic (Howat et al, 2014; Howat,2018)."
- l. 130 (Date): I found this description slightly confusing - are there 5 columns (one column for the date string, four columns for the year, month, day, and decimal date)? From the dataset, I see that it is indeed five individual columns, but the header makes it seem like there's only one column here (Date).

- o Text changed to "Date Columns: The Date column represents the acquisition time for the image used to pick the terminus for that trace. There are 4 additional columns for year, month, day and decimal date" for clarity.

- l. 135 (Satellite): How is this formatted/written?
  - o Added text "The names used are in listed in Table 2". Table 2 lists the satellite names.

- l. 144 (Scene ID): here again, it would be helpful to have more information about this. The Landsat Product ID/other satellite IDs are relatively straightforward, but what about the aerial images?
  - o If an author provided satellite ID information, then we do not change it - if someone is using TermPicks for machine learning, then they may need access to the original data. This assumes it is easier for them to request it with the original name. We added text "It includes information on the date and location for the original image. This may be listed as a file name the original author used and may store locally (Figure 10; Glacier 291) or a scene ID from a different satellite (e.g. Sentinel-1 product folder name)" for clarity. Figure 10 Glacier 291 shows an example of an original image name.

- l. 155 (Quality Flag): What does this entry look like for a given image? From the dataset, I see that it's comma-separated 2-digit strings (00, 01, 02, 03, 04, 05) - I'm not sure I would have gotten that from the description here.
  - o Added text "If there are multiple flags, they are separated by commas (Figure 10; Glacier 278)" for clarity. Figure 10 Glacier 278 shows an example of multiple flags.

- l. 170: where do the glacier centerlines come from?
  - o Text added "Centerlines are manually mapped from the MEaSUREs Greenland Ice Mapping Project (GIMP) 2000 Image Mosaic (Howat et al., 2014; Howat, 2018)."

- l. 226: how many of these picks needed manual checking?
  - o Only 220 traces were checked manually for this section. Text changed to "Traces with >500 m error between traces were manually checked for errors (220 traces)."

- l. 228: wouldn't it make more sense to compare the image (assuming it exists) against the different picks, rather than using the completeness of the metadata?

- o The method we used to compare traces between large errors in multiple authors assumes the large error is due to mislabeling the date (i.e. the trace did not appear to be from the same front on the same date as there is a large step change in the traces). The author that included the original image likely kept detailed record of what image was used and therefore is less likely to have incorrectly listed the date. As this was a very small subset of the dataset (~0.4%) we chose not to manually check each trace.
- Figure 5: I really like this figure.
  - o Thank you!
- The GEEDiT walkthrough is great - have you thought about putting it on github pages (https://pages.github.com/) so that it's more widely visible/available?
  - o GEEDiT TermPicks has been put into a repository (https://github.com/jmlea16/GEEDiT-TermPicks) documenting the walkthrough and program.

---

## Author Comment (AC2)

The manuscript from Goliber et al. collates terminus shapefile from a variety of different published studies into one dataset, complete with metadata, with the ultimate aim that the dataset could be used as training data for machine learning.

I think this is both an excellent manuscript and dataset and I enjoyed having a look through the dataset and the associated Google Earth file. I certainly recommend the publication of this manuscript in The Cryosphere. I do have a few very minor comments which the authors may wish to consider.

Thank you for the positive feedback and comments on the manuscript, and we are glad you enjoyed looking through the data. As we addressed comments, the original line numbers of the text may have changed in the final manuscript. The changed text has been noted in the responses to individual comments. Our responses are in blue below each comment.

Line 91: Why exclude glaciers with less than two authors digitizing them? What is the rationale for this?

- My text here is unclear and overly complicated. We decided to exclude glaciers with only a single trace and therefore no timeseries information. These were generally glaciers that were very small. The text has been changed to "We excluded terminus picks where only one pick was available for the glacier over all authors as well as land-terminating glaciers" for clarity.

Section 3.2: Is there a bias here, in that most of the repeated terminus picks I presume are from the later periods i.e. 2000-2020. Here the imagery is of much superior quality, which would result in a lower error. In particular most of the Landsat-1 scenes have a pretty poor geolocation accuracy and often require a manual correction, could this result in a much larger error?

- Yes, this may be the case and we do find slight difference between errors pre-2000s to post 2000s. However, much of the largest errors (>5k) are in the 2000-10 due to differences in tracing of fractured ice tongues. The figure below is for dates with more than one trace from at least 2 different authors with a Hausdorff distance of <500 m. It shows there is a slight increase in error (<200m) in the 2000s but there are also more duplicated traces during that time overall.  These are all the dates pre-removal noted in line 232.

[Figure]

Figure 9: There seems to be a large difference between the authors in this figure in the calculated retreat, but I can not distinguish any difference on the figure due to the thickness of the shapefile. Could the thickness of the shapefiles be reduced to help with this?

- Figure has been updated with new colors, reduced line thickness and opacity to help distinguish the difference.

---

## Author Comment (AC3)

Summary: The authors compiled all publicly-available Greenland marine-terminating outlet glacier positions from a wide variety of authors and performed a rigorous standardization procedure with the aim of creating a terminus trace database that could train machine learning algorithms. A description of qualitative and quantitative differences between the sources is provided, as well as a cursory review of the terminus position data coverage and estimated retreat rates relative to single datasets. The discussion focuses on recommendations for use of these data in machine learning algorithms as well as generation of additional manual terminus trace data using the updated GEEDiT tool (called GEEDiT-TermPicks).

The manuscript is easy to read and documents much-needed work. Although I hope the standardized datasets and the "ideal" approach and output format for the terminus data will advance our field, I am a bit disappointed that this manuscript did not describe any novel insights gained from the combined dataset. I assume that is the topic of another manuscript, but it would have been nice to have this manuscript go a bit beyond a dataset description.

We appreciate the constrictive feedback and positive comments on the manuscript. Based on Reviewer #3's comments, we expanded on the usefulness of the dataset for both scientific and machine learning purposes in the text, primarily by improving figure 8. While we appreciate the desire for additional analysis, the manuscript itself is meant to present a new dataset that will be widely used by the glaciology community to produce new science with estimates of errors and temporal and spatial biases present in terminus traces. Additionally, many results regarding retreat have been published by the original data providers. As we addressed comments, the original line/section numbers of the text may have changed in the final manuscript. The changed text has been noted in the responses to individual comments. Our responses are in blue below each comment.

Major Points:

1. I'm not a huge fan of the title. A think there are lots of other applications for this dataset and I think it does the dataset a disservice for the title to suggest it can only be beneficial to machine learning applications. Also, there is no demonstration how the dataset improves machine learning applications (although the authors site machine learning manuscripts focused on glacier change). Instead, I recommend something broader, like "A standardized dataset and workflow for Greenland glacier terminus positions".

   - Title changed to "TermPicks: A century of Greenland glacier terminus data for use in scientific and machine learning applications." While we do not claim that we will improve machine learning itself, the addition of the new

training data that includes image IDs will aid in improving the ability of machine learning to identify fronts in times of obstruction due to environmental factors and poor image quality (ice mélange, image saturation in early Landsat, etc.). This was an identified need to improve machine learning application by our co-authors who work on these issues. We agree with the reviewer that this data set will not only be useful for machine learning scientists. In section 2.4, we added the sentence "Including scene IDs is also useful in cases where scientists want to explore other features in the scene at the time of a terminus trace (e.g. iceberg distribution, sediment plume occurrence)" to make this more clear.

2. I appreciate that the results focus on errors and biases for individual traces, but I would also like more information on what the dataset can tell us about changes over time. This does not have to be a Greenland-wide description, but it is important to demonstrate how the combined dataset is much improved over individual datasets. There is one example figure (Figure 8) that is briefly mentioned in the discussion section as an example of the more "complete view of the change" for a glacier. It would be helpful if more examples were given, say as a series of subplots, and that some patterns in retreat rate, magnitude, or timing of changes in those metrics were presented for the broader dataset. Figure 6 gets close to doing this sort of broad overview to demonstrate merit, but doesn't adequately emphasize the value added by combining the datasets. If these sorts of metrics were presented for some of the contributing datasets as well, I think that information would really emphasize the need for coordination of efforts so that records are detailed in time but also extensive in both space and time. Right now there isn't anything that demonstrates the broad importance of the dataset you worked hard to create.

- The authors plan on publishing subsequent papers on the application of the dataset, however the goal of the manuscript is to present a combined dataset with the addition of standardized metadata and image IDs for scientists to easily use these data. One of the largest indicators of the need for coordination is not only the usefulness, but the time it takes to create these datasets. In line 50, we estimate that it took approximately 48 hours per glacier to pick all available images in the Catania and others (2018) paper. Duplication of efforts precludes scientists from working on new questions and the goal of this paper is to reduce that.
- To showcase the datasets merit further, we included subplots of individual author data in addition to the overall TermPicks dataset in figure 8 and compare the magnitude and retreat rate for a subset of authors (Moon, Fahrner, Carr, Murray) in 2000-2010. The retreat

magnitude and rates are comparable, the seasonality is only apparent when you include more data points. While the Fahrner data provides a single trace per year and the Carr and Moon data provide <1 trace per year on average to get the long-term magnitude of retreat, the lack of additional traces per year precludes the calculation of seasonality. While the record covers a shorter time, with an average of 6 traces per year for this glacier the Murray data provides enough traces per year to calculate a seasonal signal. The addition of the other authors (Korsgaard, Black, Wood) allows longer term retreat study and analysis of seasonality over the entire record.

- Updates Figure 8:

[Figure]

| Author | Start | End | Retreat magnitude (km) | Retreat rate (m/yr) | Seasonality (m) |
|---|---|---|---|---|---|
| TermPicks | 5/29/2000 | 9/21/2010 | -2.01 | -194.8 | 106 |
| Moon | 1/22/2001 | 1/28/2009 | -1.74 | -216.9 | N/A |
| Carr | 8/24/2000 | 8/13/2010 | -1.365 | -136.8 | N/A |
| Fahrner | 9/18/2000 | 9/11/2009 | -1.425 | -158.6 | N/A |
| Murray | 5/29/2000 | 9/15/2008 | -1.92 | -231.2 | 157 |

3. I'm not sure if this should be swapped in as a main figure or added as a supplemental figure, but I'd like to see heat maps or actual maps of the average temporal resolution and coverage for each glacier. You could potentially use

different symbol sizes and colors on an actual map to display those data. Right now the focus is on the number of traces for each glacier, which is important for machine learning, but the temporal resolution and coverage is much more important for someone who would want to analyze these data.

- Figures A9-11 in the Appendix demonstrate the number of traces per year for each glacier in our dataset. This shows how the temporal distribution of picks varies over each glacier. Additionally, we provide a Google Earth .kmz file in our data submission available on Zenodo that includes a Landsat coverage figure (examples shown in Figure 5) for each glacier so users can see the temporal coverage over the year for each glacier. While this only includes the Landsat data, as 70% of the dataset is Landsat, it provides a good overview of the temporal resolution and coverage for glaciers of interest.

4. In my opinion, the data formatting section should be below the metadata creation section. You mention scene IDs in the metadata creation but that comes after you already describe how you assigned IDs for datasets that did not contain that bit of metadata.

- The name of the section was changed to "Landsat image scene identifiers" and moved below "Metadata Creation" section for clarity.

Minor Comments:

- Why is the ID flag 005 but all the other flags begin with X?

  - The flag of 05 referenced in section 2.5 Landsat image scene identifiers (formally "data formatting) refers to assigning Landsat IDs to only manually-delineated traces, therefore the prefix (X) of the quality flag will be 0. If it were referring to automatic traces, it would be 1.

- Section 3.3: There needs to be more quantitative substance here. You briefly state that you observe changes in retreat rates. What are the retreat rates? See my major comment about including more of a comparison with the contributing datasets to demonstrate difference.

  - The goal of this paper is to present a dataset that can be used widely by the scientific community. Many previous studies have already published retreat (Murray et al., 2015a; Cowton et al., 2018; Wood et al., 2021) and retreat rates (Box et al., 2017; King et al., 2020) and controls on retreat (Murray et al., 2015b; Catania et al., 2018; Fried et al., 2018; Slater et al., 2019). The purpose of the retreat section is to provide a check that our

dataset does not differ greatly from any of these previous studies. We plan to publish more detailed results with our terminus dataset in upcoming publications.

**References:**

Box, J. E., & Decker, D. T. (2011). Greenland marine-terminating glacier area changes: 2000–2010. *Annals of Glaciology*, *52*(59), 91-98.

Catania, G. A., Stearns, L. A., Sutherland, D. A., Fried, M. J., Bartholomaus, T. C., Morlighem, M., ... & Nash, J. (2018). Geometric controls on tidewater glacier retreat in central western Greenland. *Journal of Geophysical Research: Earth Surface*, *123*(8), 2024-2038.

Cowton, T. R., Sole, A. J., Nienow, P. W., Slater, D. A., & Christoffersen, P. (2018). Linear response of east Greenland's tidewater glaciers to ocean/atmosphere warming. *Proceedings of the National Academy of Sciences*, *115*(31), 7907-7912.

Fried, M. J., Catania, G. A., Stearns, L. A., Sutherland, D. A., Bartholomaus, T. C., Shroyer, E., & Nash, J. (2018). Reconciling drivers of seasonal terminus advance and retreat at 13 Central West Greenland tidewater glaciers. *Journal of Geophysical Research: Earth Surface*, *123*(7), 1590-1607.

King, M. D., Howat, I. M., Candela, S. G., Noh, M. J., Jeong, S., Noël, B. P., ... & Negrete, A. (2020). Dynamic ice loss from the Greenland Ice Sheet driven by sustained glacier retreat. *Communications Earth & Environment*, *1*(1), 1-7.

Murray, T., Scharrer, K., Selmes, N., Booth, A. D., James, T. D., Bevan, S. L., ... & McGovern, J. (2015a). Extensive retreat of Greenland tidewater glaciers, 2000–2010. *Arctic, antarctic, and alpine research*, *47*(3), 427-447.

Murray, T., Selmes, N., James, T. D., Edwards, S., Martin, I., O'Farrell, T., ... & Baugé, T. (2015b). Dynamics of glacier calving at the ungrounded margin of Helheim Glacier, southeast Greenland. *Journal of Geophysical Research: Earth Surface*, *120*(6), 964-982.

Slater, D. A., Straneo, F., Felikson, D., Little, C. M., Goelzer, H., Fettweis, X., & Holte, J. (2019). Estimating Greenland tidewater glacier retreat driven by submarine melting. *The Cryosphere*, *13*(9), 2489-2509.

Wood, M., Rignot, E., Fenty, I., An, L., Bjørk, A., van den Broeke, M., ... & Zhang, H. (2021). Ocean forcing drives glacier retreat in Greenland. *Science advances*, *7*(1), eaba7282.

---

## Author Response (AR5)

Initial response to editor 12 Oct 2021

TermPicks: A century of Greenland glacier terminus data for use in machine learning applications

Authors Response to the Editor

**Comments to be addressed:**

**Add author affiliation #6, which is missing.**

- Added Affiliation [6]  University of California at Irvine, Irvine, CA, USA

**Fix year in references Mohajerani et al. (2019) and Baumhoer et al. (2019) -- they have extra digits appended.**

- Line 53: .bib file fixed so additional digits are removed.

**Add location information to Figure 2. This could be x/y or lat/lon on the axes, or simply glacier names and ID numbers in the caption (as done for Figure 9).**

- Glacier IDs added to caption of Figure 2.

**Correct the typos in all captions of Figures A*: GEEDiT misspelled as GEEiT. Was this intentional?**

- All typos of 'GEEiT' corrected to GEEDiT in appendix

**The caption of Table 2 says "Satellites", but the table also includes air and ground photos from the 1930s-1970s.**

- Cation and column label changed from "Satellites" to "Sources"

**I could not find a url or way to access GEEDiT-TermPicks.**

- Line 314: Added Link to GEEDiT TermPicks
- Appendix C: Added Link to GEEDiT TermPicks

**A brief description of the box method (line 76) would be helpful. You currently cite Lea et al. (2014) for a full description, but a single sentence summary of the box method here would go a long way for people with some, but not extensive, knowledge of terminus picking.**

- Line 78: Added single sentence summary of 'box methods'

**Clarification of an apparent inconsistency that the dataset goes back to 1916 for some glaciers (lines 181, 303), yet in Table 2, the earliest dates are 1931 (Bjork) or 1948 (Hill).**

- Table 2: Added missing citation for Hill et al., 2017 that was not provided by original author. The data set is unchanged as the author tag in the data is still "Hill."

**Author response to "Editor decision: Publish subject to revisions"**

At the recommendation of the editor, we included two new results that focus on the demonstration of the usefulness of TermPicks that was missing in individual datasets. Section 2.6 in the methods describes how we estimated seasonality and sinuosity from the TermPicks dataset. Estimating seasonality showcases that with TermPicks, seasonality can be determined for a far greater number of glaciers over a longer time span than any author alone. This is shown in Figure 7 and described in Results (Line 205) and Discussion (Line 281). We also showcase the need for the fullwidth terminus trace instead of a centerline-only trace through the calculation of sinuosity for glaciers shown in Figure 8 and described in Results (Line 213) and Discussion (Line 285). As we included two new figures, one of which showing seasonality, we did not include the updated figure 11 (Originally Figure 8) that we included in our response to reviewer # 3. Our new figure 7 shows how the inclusion of additional data estimates seasonality in a more sufficient way than our previous figure for many glaciers. Inclusion of these results led to some restructuring of the methods, results, and discussion section.

The Track-changes document is not in the proper Copernicus format as my compiler time-out (Overleaf) when attempting to use a LaTeX Diff file using a Copernicus document class. All relevant changes should be there, except for changes made to the Authors list (adding town, Country information).

TermPicks Referee #1

In this manuscript, the authors have described a dataset of manually digitized terminus positions for outlet glaciers of the Greenland ice sheet compiled from previously-published datasets, in order to provide a consistently-formatted training dataset for future machine learning applications. This is an excellent and timely undertaking that highlights the power of collaborative efforts.

On the whole, the manuscript does a good job describing the issues involved in combining "input" datasets from multiple authors, as well as describing the "ouptut" dataset, and even manages to show an example application of combining data sources. Accordingly, I only have a few minor comments/suggestions to make on the manuscript. The bulk of my comments/suggestions have to do with the description of the metadata - I think a Table with a few different example entries would help clarify this for a reader.

We appreciate the constructive and positive feedback on the manuscript. We address the comments bellow, but also included a new figure (10) of the metadata of three glaciers to better clarify the structure. As we addressed comments, the original line numbers of the text may have changed in the final manuscript. The changed text has been noted in the responses to individual comments. Our responses are in blue below each comment.

Comments to Address:

- l. 10: is this the mean (± standard deviation)?
    - Yes. This was changed to "The TermPicks data set includes 39,060 individual terminus traces for 278 glaciers with a mean of 136±190 and median of 93 of traces per glacier" to be more clear. The SD is higher than the mean due to the high variation of picks between certain glaciers.
- l. 52: check that months are removed from the reference dates
    - These have been checked and have removed from the text.
- l. 104: is the Howat reference here for the MODIS image?
    - The incorrect MEaSUREs image was being cited here. It has been changed to "MEaSUREs Greenland Ice Mapping Project (GIMP) 2000 Image Mosaic (Howat et al, 2014; Howat,2018)."

- l. 130 (Date): I found this description slightly confusing - are there 5 columns (one column for the date string, four columns for the year, month, day, and decimal date)? From the dataset, I see that it is indeed five individual columns, but the header makes it seem like there's only one column here (Date).

  o Text changed to "Date Columns: The Date column represents the acquisition time for the image used to pick the terminus for that trace. There are 4 additional columns for year, month, day and decimal date" for clarity.

- l. 135 (Satellite): How is this formatted/written?

  o Added text "The names used are in listed in Table 2". Table 2 lists the satellite names.

- l. 144 (Scene ID): here again, it would be helpful to have more information about this. The Landsat Product ID/other satellite IDs are relatively straightforward, but what about the aerial images?

  o If an author provided satellite ID information, then we do not change it - if someone is using TermPicks for machine learning, then they may need access to the original data. This assumes it is easier for them to request it with the original name. We added text "It includes information on the date and location for the original image. This may be listed as a file name the original author used and may store locally (Figure 10; Glacier 291) or a scene ID from a different satellite (e.g. Sentinel-1 product folder name)" for clarity. Figure 10 Glacier 291 shows an example of an original image name.

- l. 155 (Quality Flag): What does this entry look like for a given image? From the dataset, I see that it's comma-separated 2-digit strings (00, 01, 02, 03, 04, 05) - I'm not sure I would have gotten that from the description here.

  o Added text "If there are multiple flags, they are separated by commas (Figure 10; Glacier 278)" for clarity. Figure 10 Glacier 278 shows an example of multiple flags.

- l. 170: where do the glacier centerlines come from?

  o Text added "Centerlines are manually mapped from the MEaSUREs Greenland Ice Mapping Project (GIMP) 2000 Image Mosaic (Howat et al., 2014; Howat, 2018)."

- l. 226: how many of these picks needed manual checking?

- o Only 220 traces were checked manually for this section. Text changed to "Traces with >500 m error between traces were manually checked for errors (220 traces)."
- l. 228: wouldn't it make more sense to compare the image (assuming it exists) against the different picks, rather than using the completeness of the metadata?
  - o The method we used to compare traces between large errors in multiple authors assumes the large error is due to mislabeling the date (i.e. the trace did not appear to be from the same front on the same date as there is a large step change in the traces). The author that included the original image likely kept detailed record of what image was used and therefore is less likely to have incorrectly listed the date. As this was a very small subset of the dataset (~0.4%) we chose not to manually check each trace.
- Figure 5: I really like this figure.
  - o Thank you!
- The GEEDiT walkthrough is great - have you thought about putting it on github pages (https://pages.github.com/) so that it's more widely visible/available?
  - o GEEDiT TermPicks has been put into a repository (https://github.com/jmlea16/GEEDiT-TermPicks) documenting the walkthrough and program.

TermPicks Referee #2

The manuscript from Goliber et al. collates terminus shapefile from a variety of different published studies into one dataset, complete with metadata, with the ultimate aim that the dataset could be used as training data for machine learning.

I think this is both an excellent manuscript and dataset and I enjoyed having a look through the dataset and the associated Google Earth file. I certainly recommend the publication of this manuscript in The Cryosphere. I do have a few very minor comments which the authors may wish to consider.

Thank you for the positive feedback and comments on the manuscript, and we are glad you enjoyed looking through the data. As we addressed comments, the original line numbers of the text may have changed in the final manuscript. The changed text has been noted in the responses to individual comments. Our responses are in blue below each comment.

Line 91: Why exclude glaciers with less than two authors digitizing them? What is the rationale for this?

- My text here is unclear and overly complicated. We decided to exclude glaciers with only a single trace and therefore no timeseries information. These were generally glaciers that were very small. The text has been changed to "We excluded terminus picks where only one pick was available for the glacier over all authors as well as land-terminating glaciers" for clarity.

Section 3.2: Is there a bias here, in that most of the repeated terminus picks I presume are from the later periods i.e. 2000-2020. Here the imagery is of much superior quality, which would result in a lower error. In particular most of the Landsat-1 scenes have a pretty poor geolocation accuracy and often require a manual correction, could this result in a much larger error?

- Yes, this may be the case and we do find slight difference between errors pre-2000s to post 2000s. However, much of the largest errors (>5k) are in the 2000-10 due to differences in tracing of fractured ice tongues. The figure below is for dates with more than one trace from at least 2 different authors with a Hausdorff distance of <500 m. It shows there is a slight increase in error (<200m) in the 2000s but there are also more

[Figure]

duplicated traces during that time overall.  These are all the dates pre-removal noted in line 232.

Figure 9: There seems to be a large difference between the authors in this figure in the calculated retreat, but I can not distinguish any difference on the figure due to the thickness of the shapefile. Could the thickness of the shapefiles be reduced to help with this?

- Figure has been updated with new colors, reduced line thickness and opacity to help distinguish the difference.

TermPicks Referee #3

Summary: The authors compiled all publicly-available Greenland marine-terminating outlet glacier positions from a wide variety of authors and performed a rigorous standardization procedure with the aim of creating a terminus trace database that could train machine learning algorithms. A description of qualitative and quantitative differences between the sources is provided, as well as a cursory review of the terminus position data coverage and estimated retreat rates relative to single datasets. The discussion focuses on recommendations for use of these data in machine learning algorithms as well as generation of additional manual terminus trace data using the updated GEEDiT tool (called GEEDiT-TermPicks).

The manuscript is easy to read and documents much-needed work. Although I hope the standardized datasets and the "ideal" approach and output format for the terminus data will advance our field, I am a bit disappointed that this manuscript did not describe any novel insights gained from the combined dataset. I assume that is the topic of another manuscript, but it would have been nice to have this manuscript go a bit beyond a dataset description.

We appreciate the constrictive feedback and positive comments on the manuscript. Based on Reviewer #3's comments, we expanded on the usefulness of the dataset for both scientific and machine learning purposes in the text, primarily by improving figure 8. While we appreciate the desire for additional analysis, the manuscript itself is meant to present a new dataset that will be widely used by the glaciology community to produce new science with estimates of errors and temporal and spatial biases present in terminus traces. Additionally, many results regarding retreat have been published by the original data providers. As we addressed comments, the original line/section numbers of the text may have changed in the final manuscript. The changed text has been noted in the responses to individual comments. Our responses are in blue below each comment.

Major Points:

1. I'm not a huge fan of the title. A think there are lots of other applications for this dataset and I think it does the dataset a disservice for the title to suggest it can only be beneficial to machine learning applications. Also, there is no demonstration how the dataset improves machine learning applications (although the authors site machine learning manuscripts focused on glacier

change). Instead, I recommend something broader, like "A standardized dataset and workflow for Greenland glacier terminus positions".

- Title changed to "TermPicks: A century of Greenland glacier terminus data for use in scientific and machine learning applications." While we do not claim that we will improve machine learning itself, the addition of the new training data that includes image IDs will aid in improving the ability of machine learning to identify fronts in times of obstruction due to environmental factors and poor image quality (ice mélange, image saturation in early Landsat, etc.). This was an identified need to improve machine learning application by our co-authors who work on these issues. We agree with the reviewer that this data set will not only be useful for machine learning scientists. In section 2.4, we added the sentence "Including scene IDs is also useful in cases where scientists want to explore other features in the scene at the time of a terminus trace (e.g. iceberg distribution, sediment plume occurrence)" to make this more clear.

2. I appreciate that the results focus on errors and biases for individual traces, but I would also like more information on what the dataset can tell us about changes over time. This does not have to be a Greenland-wide description, but it is important to demonstrate how the combined dataset is much improved over individual datasets. There is one example figure (Figure 8) that is briefly mentioned in the discussion section as an example of the more "complete view of the change" for a glacier. It would be helpful if more examples were given, say as a series of subplots, and that some patterns in retreat rate, magnitude, or timing of changes in those metrics were presented for the broader dataset. Figure 6 gets close to doing this sort of broad overview to demonstrate merit, but doesn't adequately emphasize the value added by combining the datasets. If these sorts of metrics were presented for some of the contributing datasets as well, I think that information would really emphasize the need for coordination of efforts so that records are detailed in time but also extensive in both space and time. Right now there isn't anything that demonstrates the broad importance of the dataset you worked hard to create.

- The authors plan on publishing subsequent papers on the application of the dataset, however the goal of the manuscript is to present a combined dataset with the addition of standardized metadata and image IDs for scientists to easily use these data. One of the largest indicators of the need for coordination is not only the usefulness, but

the time it takes to create these datasets. In line 50, we estimate that it took approximately 48 hours per glacier to pick all available images in the Catania and others (2018) paper. Duplication of efforts precludes scientists from working on new questions and the goal of this paper is to reduce that.

- To showcase the datasets merit further, we included subplots of individual author data in addition to the overall TermPicks dataset in figure 8 and compare the magnitude and retreat rate for a subset of authors (Moon, Fahrner, Carr, Murray) in 2000-2010. The retreat magnitude and rates are comparable, the seasonality is only apparent when you include more data points. While the Fahrner data provides a single trace per year and the Carr and Moon data provide <1 trace per year on average to get the long-term magnitude of retreat, the lack of additional traces per year precludes the calculation of seasonality. While the record covers a shorter time, with an average of 6 traces per year for this glacier the Murray data provides enough traces per year to calculate a seasonal signal. The addition of the other authors (Korsgaard, Black, Wood) allows longer term retreat study and analysis of seasonality over the entire record.

- Updates Figure 8:

[Figure]

| Author | Start | End | Retreat magnitude (km) | Retreat rate (m/yr) | Seasonality (m) |
|---|---|---|---|---|---|
| TermPicks | 5/29/2000 | 9/21/2010 | -2.01 | -194.8 | 106 |
| Moon | 1/22/2001 | 1/28/2009 | -1.74 | -216.9 | N/A |
| Carr | 8/24/2000 | 8/13/2010 | -1.365 | -136.8 | N/A |
| Fahrner | 9/18/2000 | 9/11/2009 | -1.425 | -158.6 | N/A |
| Murray | 5/29/2000 | 9/15/2008 | -1.92 | -231.2 | 157 |

3. I'm not sure if this should be swapped in as a main figure or added as a supplemental figure, but I'd like to see heat maps or actual maps of the average temporal resolution and coverage for each glacier. You could potentially use different symbol sizes and colors on an actual map to display those data. Right now the focus is on the number of traces for each glacier, which is important for machine learning, but the temporal resolution and coverage is much more important for someone who would want to analyze these data.

- Figures A9-11 in the Appendix demonstrate the number of traces per year for each glacier in our dataset. This shows how the temporal distribution of picks varies over each glacier. Additionally, we provide a Google Earth .kmz file in our data submission available on Zenodo that includes a Landsat coverage figure (examples shown in Figure 5) for each glacier so users can see the temporal coverage over the year for

each glacier. While this only includes the Landsat data, as 70% of the dataset is Landsat, it provides a good overview of the temporal resolution and coverage for glaciers of interest.

4. In my opinion, the data formatting section should be below the metadata creation section. You mention scene IDs in the metadata creation but that comes after you already describe how you assigned IDs for datasets that did not contain that bit of metadata.

- The name of the section was changed to "Landsat image scene identifiers" and moved below "Metadata Creation" section for clarity.

Minor Comments:

- Why is the ID flag 005 but all the other flags begin with X?
  - The flag of 05 referenced in section 2.5 Landsat image scene identifiers (formally "data formatting) refers to assigning Landsat IDs to only manually-delineated traces, therefore the prefix (X) of the quality flag will be 0. If it were referring to automatic traces, it would be 1.
- Section 3.3: There needs to be more quantitative substance here. You briefly state that you observe changes in retreat rates. What are the retreat rates? See my major comment about including more of a comparison with the contributing datasets to demonstrate difference.
  - The goal of this paper is to present a dataset that can be used widely by the scientific community. Many previous studies have already published retreat (Murray et al., 2015a; Cowton et al., 2018; Wood et al., 2021) and retreat rates (Box et al., 2017; King et al., 2020) and controls on retreat (Murray et al., 2015b; Catania et al., 2018; Fried et al., 2018; Slater et al., 2019). The purpose of the retreat section is to provide a check that our dataset does not differ greatly from any of these previous studies. We plan to publish more detailed results with our terminus dataset in upcoming publications.

**References:**

Box, J. E., & Decker, D. T. (2011). Greenland marine-terminating glacier area changes: 2000–2010. *Annals of Glaciology*, *52*(59), 91-98.

Catania, G. A., Stearns, L. A., Sutherland, D. A., Fried, M. J., Bartholomaus, T. C., Morlighem, M., ... & Nash, J. (2018). Geometric controls on tidewater glacier retreat in central western Greenland. *Journal of Geophysical Research: Earth Surface*, *123*(8), 2024-2038.

Cowton, T. R., Sole, A. J., Nienow, P. W., Slater, D. A., & Christoffersen, P. (2018). Linear response of east Greenland's tidewater glaciers to ocean/atmosphere warming. *Proceedings of the National Academy of Sciences*, *115*(31), 7907-7912.

Fried, M. J., Catania, G. A., Stearns, L. A., Sutherland, D. A., Bartholomaus, T. C., Shroyer, E., & Nash, J. (2018). Reconciling drivers of seasonal terminus advance and retreat at 13 Central West Greenland tidewater glaciers. *Journal of Geophysical Research: Earth Surface*, *123*(7), 1590-1607.

King, M. D., Howat, I. M., Candela, S. G., Noh, M. J., Jeong, S., Noël, B. P., ... & Negrete, A. (2020). Dynamic ice loss from the Greenland Ice Sheet driven by sustained glacier retreat. *Communications Earth & Environment*, *1*(1), 1-7.

Murray, T., Scharrer, K., Selmes, N., Booth, A. D., James, T. D., Bevan, S. L., ... & McGovern, J. (2015a). Extensive retreat of Greenland tidewater glaciers, 2000–2010. *Arctic, antarctic, and alpine research*, *47*(3), 427-447.

Murray, T., Selmes, N., James, T. D., Edwards, S., Martin, I., O'Farrell, T., ... & Baugé, T. (2015b). Dynamics of glacier calving at the ungrounded margin of Helheim Glacier, southeast Greenland. *Journal of Geophysical Research: Earth Surface*, *120*(6), 964-982.

Slater, D. A., Straneo, F., Felikson, D., Little, C. M., Goelzer, H., Fettweis, X., & Holte, J. (2019). Estimating Greenland tidewater glacier retreat driven by submarine melting. *The Cryosphere*, *13*(9), 2489-2509.

Wood, M., Rignot, E., Fenty, I., An, L., Bjørk, A., van den Broeke, M., ... & Zhang, H. (2021). Ocean forcing drives glacier retreat in Greenland. *Science advances*, *7*(1), eaba7282.

Response to editor 15 Jun 2022

Dear Sophie Goliber and co-authors,

Thank you for your replies to the reviewers and the corresponding changes you made to the manuscript. The addition of two results that illustrate the power of the dataset (seasonality and sinuosity, Figures 7 and 8) strengthen the manuscript.

I have a number of comments regarding the organization of the manuscript and the precision of the descriptions that need to be addressed before this can be published in The Cryosphere. First, a number of findings from the TermPicks dataset are currently presented in the Discussion section. These include results on terminus sinuosity and on the effects of lateral end points of termini on fjord-mean terminus position. These paragraphs should be moved to the Results section. Second, the clarity of some of the added text is less good than in the original manuscript. I copied one example below. I suggest that all changes (including to this and future versions) be reviewed again by the primary and supporting authors to ensure precision and smoothness.
"We estimate seasonality for years in which there are terminus picks in at least three unique months to illustrate the density of the data set."

Thank you for your comments and helpful review of our new addition to the manuscript. We have reviewed and restructured the new sections (seasonality and sinuosity) in the methods, results, and discussion sections for clarity. The largest change is in the sinuosity figure (#10, previously #8). We have also proofread and edited for clarity in the overall text, including editing the figure order. Our responses to individual comments can be found below.

Specific items that need addressing:

Line 146: Center X and Y: In the dataset called TermPicks+CALFIN_V2, X and Y are identical (both appear to be longitude).

Fixed and updated.

Line 151: Specify whether X=0 or 1 is automatically or manually created.

The prefix is assigned based on the method that the trace was created. All TermPicks data is from manually-digitized data, therefore it has a prefix of 0. Any machine-generated traces (i.e. CALFIN) have a qualify flag with a prefix of 1. In the future, if machine-generated datasets would like to be used in conjunction with TermPicks, they would have a qualify flag with a prefix of 1. To clarify this, we added the following text:

"We assign a prefix 'X' for all data defining if the trace was created automatically or manually with X=0 for TermPicks data and X=1 for CALFIN data, or any machine-generated terminus traces that may be included in the future"

And edited table 3:

| Flag Code | Issue |
|---|---|
| X = 0 | Manually-digitized trace |
| X = 1 | Machine-generated trace |
| X0 | No issues |
| X1 | Trace uncertainty due to environment or image issues (clouds, shadows, missing data, etc.) |
| X2 | Supplemented trace |
| X3 | Landsat 7 SLC off |
| X4 | Incomplete/Box Method |
| X5 | Automatically assigned scene ID |

**Table 3.** Flags assigned to output terminus trace data created in conjunction with CALFIN Cheng et al. (2020). All data in the TermPicks dataset has the prefix of X = 0.

Line 244: Rephrase to remove the nested parentheses.

Removed. Rephrased to "As a metric of error between data sets, we calculated the Hausdorff distance (commonly used in pattern recognition), the greatest minimum distance between two lines (Huttenlocher et al., 1993)."

Paragraph beginning on line 284: This text should belong in Results rather than Discussion. It should also point to Figure 8, which illstrates these results.

We have restructured this paragraph and split the appropriate sections into the methods, results, and discussion.

Line 289: "high sinuosity" - This statement is subjective and needs to be more

objective. Figure 8 shows that the sinuosity of Glacier #291 (~1.4) is pretty comparable to that of the only other sinuosity presented (Glacier #288, ~1.3-1.4). Perhaps the variance of sinuosity across glaciers is this low (~0.1); if so, that should be explained. It is simply too unclear from what is currently presented.

We were primarily focused on the change in sinuosity in time may reveal differences in processes effecting a single glacier through an example of a glacier that retreats (288) and one that remains stable (291). While the mean sinuosity over the record is comparable, the variability throughout the time series is notable. Sinuosity values generally range between 1 and 3 (Schumm, 1985), but we do not expect glacier termini to exceed a sinuosity of 2 (i.e. the terminus will be less than twice the length of the distance across the fjord), therefore we argue the range in sinuosity over time are notable. We believe the new figure shows this more clearly. As 288 begins retreating over a prograde slope, it becomes more sinuous. The sinuosity decreases as retreat rate increases (~2010) through an over-deepening in the bed. Conversely, glacier 291 does not show much change over the record.

As the sinuosity is a simple ratio of the length of the trace to the length between endpoints, it quantifies how much the terminus deviates from a straight line. Therefore, a curved terminus will also have a higher sinuosity compared to a crenulated one, so it is an imperfect metric. However, combined with other metrics such as curvature or skewness, it is useful for describing the shape of a terminus, but this analysis is out of scope for this paper and will be explored by the authors in future work.

Figure 8: This relates to the above comment. Oddly, the sinuosity for Glacier #291 in the 2020s (yellow) does not appear to be different from the rest of the record in the top plot, but in the bottom panel (map), the 2020s terminus looks more incised or protruded (and therefore more sinuous) than the rest of record shown in the top plot (1990-2020, roughly blue-green through yellow). Is this an error in analysis, an effect of the 4-year smoothing, or an artifact of the way the data are presented?

As we now present the unsmoothed data and do not have a large difference in the 2020s, we do not think this is a problem with smoothing. The terminus may appear to be more crenulated, however the overall shape of the terminus is also important. Because of how sinuosity is calculated, a smooth but very convex terminus may have a similar sinuosity to a relatively straight, but highly crenulated terminus. This has been clarified in the text.

New figure:

[Figure]

**Figure 10.** A: Terminus change between 1990-2020 colored by sinuosity for Glacier 288 (Sermeq Silarleq). The dashed grey line is the start of progressive retreat as defined in Catania et al. (2018). B: Corresponding map-view terminus traces For Glacier 288 with every 5th trace colored by sinuosity. C: Terminus change between 1990-2020 colored by sinuosity for Glacier 291. D: Corresponding map-view terminus traces for Glacier 291 (Kangerdlugssup Sermerssua) with every 5th trace colored by sinuosity. The base map in B and D is the bed from BedMachine (Morlighem et al., 2017). The black pixels in B are errors, however they do not impact the overall interpretation of the bed. The bed scalebar applies to both B and D. The white arrows indicate glacier flow direction. The red star on the inset map is the location of the glacier on the Greenland Ice Sheet.

- Sinuosity data since 1973 appear on the map (it looks very sinuous, with a deeply incised or protruded centerline, in the 1970s) but not on the time series. Why not?

We only included sinuosity from 1990's onward because there are far more traces after the 1990s than earlier. This has been clarified in the text. Additionally, we were primarily focused on showcasing the change in sinuosity over time for glacier 288 as it retreats compared to the relatively consistent sinuosity for 291. We have edited the figure to show the unsmoothed retreat colorized by the unsmoothed sinuosity. The map-view figures how every 5th terminus trace. This shows the change in sinuosity in glacier 288 more clearly.

- It is not apparent from the map panels which directions these glaciers are flowing. It could be inferred for Glacier #288 (not #291), but an arrow label would go a long way here.

Added white arrows to indicate flow direction.

- Top panels are labeled "Retreat", with mostly negative numbers for the glacier that is in clear retreat (#288). This is inconsistent. Rename the label (recommended) or flip the scale.

Changed to "Terminus change (m)"

- The names of these glaciers appear in the text (Line 215-216), so they should also appear on the figures or at least in the caption.

Added the names to the caption.

- The overdeepened bed referred to on the text (line 292) is not discernible on the figure, as the terminus lines obscure any subtlety in the darkness of the DEM.

Change color scale to -600m-100m to increase contrast and plot every 5$^{th}$ terminus trace.

Line 291: The time series in Figure 8 for Glacier #288 doesn't really show an "after retreat" phase. It is basically retreating the whole time.

We added a grey dashed line indicating the time period where retreat begins. The glacier advances slightly, then begins retreating. The timing of progressive retreat onset is also calculated in Catania et al., 2018 as 1998.3, which we use as the marker in our plot.

Paragraph beginning on line 294: This text should belong in Results rather than Discussion.

Moved.

Line 377, 381: "TermPicks"?

Fixed.

Line 382-382:
- The TP+CALFIN_v2 X,Y error needs correcting (see previous comment).
- Glacier 291.csv is sized 0 bytes (upload fail?).

Fixed and updated in Zenodo.

Figure 6 and caption: The panels show mostly 11-year periods with one 21-year period, but this is not what the caption says (20- and 30-year periods).

Added " 's " to each date to indicate that it is an average position over 10 years for each decade. For clarity, caption edited to "For each panel, the entire decade of traces were averaged to produce an average position for that decade. The 1940/1950s are an average over both decades as there are fewer traces available in the 1950s. Then the average position is differenced from the previous decade."

Figure 7 and caption:
- Please provide names for these three glaciers, and indicate in the caption that the top left panel shows the locations of these glaciers.

Fixed and updated.

- Glacier #116 has retreat in meters; this is likely an error / typo.
Fixed and updated.
Figure 8 and caption: See comments above.

Throughout text: Choose to use the # sign for glacier number either consistently or not at all. Currently, it is mostly omitted in the main text, then mostly used in the captions and supplement.
Fixed and updated.

Once these revisions have been completed, I will be happy to review the submission again for potential final publication in The Cryosphere.

Thank you for reviewing our revised manuscript. We have made the following changed based on your suggestions:

I have only a few minor recommendations. I found at least one instance of tense-switching in the methods section. Take care when checking this section one last time.

We have re-read the methods and fixed cases of tense-switching.

I really like map figures 7-8. However, the captions for these figures should be slightly revised. For Figure 7, it should be stated whether negative values indicate retreat and positive indicate advance as traditionally interpreted. Also, I am not sure I follow what you mean with "The the average position is differenced from the previous decade."

The caption now reads:

Decadal retreat patterns for available TermPicks data using the Centerline method. For each panel, the entire decade of traces were averaged to produce an average position for that decade. The 1940/1950s are an average over both decades as there are fewer traces available in the 1950s. Then the average position for the decade is differenced from average position of the previous decade. The size correlates to magnitude of terminus change, while red (negative) indicates retreat and blue (positive) indicates advance.

For Figure 8, the caption should be reorganized to lead with the statement that those glaciers include terminus delineations for at least 3 months, which is the minimum required to resolve seasonality in terminus position and/or terminus characteristics. Also, it needs to be stated here or in the text if these maps show glaciers with at least one year of seasonally-resolved terminus delineations or some other threshold number of years.

The caption now reads:

Locations of glaciers that include terminus delineations for at least three unique months, which is the minimum number of traces required to resolve seasonality, for the entire TermPicks data set and a subset of authors. The size of the blue circle indicates how many years there are enough traces to resolve seasonality, ranging between a single year to up to 40 years.

Response to editor   07 Jul 2022

Thank you for your comments. They have improved the manuscript. We have made the following changes:

1. Return the changed verb tense in Sect. 2.2 to the present tense

Changed back to present tense.

2. Move the beginning of Sect. 3.3 to a new sub-section in Methods

Moved to a new section named 'Error Estimates'.

3. Improve two instances of unclear sentences
Changed text to:

Line 291: Bunce and Cheng will show a higher retreat compared to ESA because the Interpolation method accounts for the entire width of the glacier. Therefore the mean positions of the Bunce and Cheng traces will be further up-glacier as they do not include the lateral tails seen in the ESA trace.

Line 368: We find biases in terms of data coverage with well-studied glaciers with high coverage of terminus trace data, and other glaciers devoid of consistent coverage, showcasing the need for further manual and machine learning efforts to provide terminus data.

4. Change all British English "-ise" spellings to be consistent with the American English "-ize" in the rest of the manuscript (I believe I highlighted all instances)

Changed in text.

**Final changes 14 Jul 2022**

Changed appendix to supplementary material.